# Daytime and nighttime aerosol soluble iron formation in clean and slightly-polluted moisture air in a coastal city in eastern China

Wenshuai Li[1,2], Yuxuan Qi[1,2], Yingchen Liu[1,2], Guanru Wu[1,2], Yanjing Zhang[1,2], Jinhui Shi[3], Wenjun Qu[1,2], Lifang Sheng[1,2], Wencai Wang[1,2], Daizhou Zhang[4], Yang Zhou[1,2]

[1]Frontier Science Center for Deep Ocean Multispheres and Earth System (FDOMES) and Physical Oceanography Laboratory, Ocean University of China, Qingdao 266100, China.

[2]College of Oceanic and Atmospheric Sciences, Ocean University of China, Qingdao 266100, China.

[3]College of Environmental Science and Engineering, Ocean University of China, Qingdao 266100, China.

[4]Faculty of Environmental and Symbiotic Sciences, Prefectural University of Kumamoto, Kumamoto 862-8502, Japan.

*Correspondence to: Daizhou Zhang (dzzhang@pu-kumamoto.ac.jp) and Yang Zhou (yangzhou@ouc.edu.cn)*

**Abstract.** Photocatalytic reactions during the daytime, alongside aqueous-phase reactions occurring during both daytime and nighttime, are identified as the two primary processes facilitating the conversion of aerosol iron (Fe) from the insoluble state to the soluble state within the atmospheric environment. This study investigated the levels of total Fe ($Fe_T$) and soluble Fe ($Fe_S$) in $PM_{2.5}$ samples collected during daytime and nighttime in Qingdao, a coastal city in eastern China, evaluating the distinctive roles of these two pathways in enhancing aerosol Fe solubility (%$Fe_S$, defined as the ratio of $Fe_S$ to $Fe_T$). Under clean and humid conditions, characterized by prevailing sea breezes and a relative humidity (RH) typically above 80%, an average daytime %$Fe_S$ of 8.7% was observed, which systematically exceeded the nighttime %$Fe_S$ (6.3%). Photochemical conversions involving oxalate contributed to the higher %$Fe_S$ observed during daytime. Conversely, in scenarios where air masses originated from inland areas and exhibited slightly polluted, daytime %$Fe_S$ (3.7%) was noted to be lower than the nighttime %$Fe_S$ (5.8%). This discrepancy was attributable to the variations in RH, with nighttime RH averaging around 77%, conducive to the more efficient generation of acidic compounds, thereby accelerating $Fe_S$ production compared to the daytime, when RH was only about 62%. Furthermore, the oxidation rates of sulfur (SOR) displayed a strong correlation with RH, particularly when RH fell below 75%. A 10% increase in RH corresponded to a 7.6% rise in SOR, which served as the primary driver of the higher aerosol acidity and %$Fe_S$ at night. These findings highlight the RH-dependent activation of aqueous-phase reactions and the augmentation of daytime photocatalysis in the formation of $Fe_S$ in the coastal moisture atmosphere.

## 1 Introduction

Iron (Fe) plays a pivotal role as a micronutrient in marine ecosystems, being a critical component of atmospheric aerosol particles (Martin et al., 1994). Its deposition in high-nitrate, low-chlorophyll (HNLC) regions can trigger phytoplankton bloom, thus enhancing atmospheric carbon absorption and fixation in seawater (Watson et al., 1994; Watson and Lefévre, 1999; Toner, 2023). Notably, only the soluble fraction of Fe ($Fe_S$) in aerosols, referred to as bioavailable Fe, is accessible to phytoplankton (Zhuang et al., 1992; Sugie et al., 2013; Li et al., 2017). The proportion of $Fe_S$ to the total aerosol Fe ($Fe_T$), i.e., the aerosol Fe solubility (%$Fe_S$), is influenced by the aerosols' sources and the chemical conversion of Fe from insoluble forms to soluble forms in the atmosphere. %$Fe_S$ in fresh dust particles is typically below 1%, yet can exceed 10% in aerosols derived from combustion processes, such as fly ash from coal and oil combustion (Oakes et al., 2012; Shi et al., 2012; Wang et al., 2015; Li et al., 2022). The %$Fe_S$ in primary particles can significantly increase due to atmospheric processes, primarily through aerosol acidification via aqueous-phase reactions or photochemical conversions of precursors of acidic species (Solmon et al., 2009; Shi et al., 2015; Li et al., 2017; Hettiarachchi et al., 2019), affecting the deposition flux of aerosol $Fe_S$ over the open ocean (Chen and Siefert, 2004; Shi et al., 2013; Yang et al., 2020).

Solar radiation and ambient humidity are two key meteorological factors that greatly influence the processes of aerosol acidification. Solar irradiation induces photochemical reactions during daytime, leading to the formation of free radicals and accelerating the production of acidic species within aerosols, thereby facilitating Fe dissolution (Chen and Grassian, 2013; Liu et al., 2021b). Studies such as Fu et al. (2010) have demonstrated increased $Fe_S$ in dust samples exposed to light in the HCl solution. Furthermore, daytime photolysis of Fe-organic complexes is another pathway for $Fe_S$ formation, contributing to increased %$Fe_S$ (Weller et al., 2014; Zhang et al., 2019; Zhou et al., 2020). For example, Zhou et al. (2020) and Zhang et al. (2019) reported that photolysis of oxalate-Fe(III) complex can result in the degradation of oxalate, enhancing Fe dissolution in aerosol particles during daytime. These mechanisms have been supported by laboratory experiments and model simulations (Zhu et al., 1993; Chen and Grassian, 2013; Sorooshian et al., 2013; Pang et al., 2019; Li et al., 2021). In contrast, high ambient relative humidity (RH) can facilitate the heterogeneous/liquid phase formation of sulfate and nitrate during nighttime, increasing aerosol acidity and promoting acids-associated Fe dissolution (Liu et al., 2020; Pye et al., 2020; Wong et al., 2020). Studies like Zhang et al. (2022) observed enhanced %$Fe_S$ (>1%) at high RH levels (>60%) in winter, while Zhu et al. (2020) highlighted the greater impact of $SO_4^{2-}$ and $NO_3^-$ on %$Fe_S$ at RH above 50%. Shi et al.

(2020) noted efficient $Fe_S$ formation under foggy conditions, where $SO_4^{2-}$ and $NO_3^-$ concentrations were high due
to the absorption of precursor gases on wet particle surfaces, facilitating further water vapor absorption, and $Fe_S$
increase.
Aqueous-phase processes can occur during both daytime and nighttime, given adequate moisture. The formation
of $Fe_S$ results from the interplay between photochemistry and aqueous chemistry during daytime, whereas it relies
solely on aqueous chemistry at night. The synergistic mechanisms and their individual contributions to $Fe_S$
formation remain partially understood.
To elucidate the roles of aqueous-phase and photochemical reactions on $Fe_S$ formation, we collected $PM_{2.5}$ samples
during daytime and nighttime, separately, in a Chinese coastal city (Qingdao). Positioned under the westerlies of
the Northern Hemisphere, Qingdao acts as a primary conduit for East Asian terrestrial aerosols to the Northwestern
Pacific. Our research focuses on ascertaining the $\%Fe_S$ enhancement under clean and slightly-polluted air
conditions, reflecting typical coastal air quality. The primary goal is to delineate the contributions of aqueous-
phase reactions and photochemical processes to $\%Fe_S$ enhancement, thereby elucidating the dynamics of Fe
dissolution within the atmospheric chemistry of coastal areas.
**2 Methodology and materials**
**2.1 Sample collection and classification**
The observation was carried out on the following dates: April 24[th] to May 27[th], 2017; March 28[th] to April 30[th],
2018; and May 22[nd] to 28[th], 2018. Two high-volume $PM_{2.5}$ samplers (TISCH, TE-6070BLX-2.5, USA) were
applied to collect $PM_{2.5}$ onto quartz microfiber filters (QM-A, PALL) and Whatman® 41 filters, respectively, on
the roof of Baguanshan Atmospheric Research Observatory (BARO, 36°03' N, 120°20' E, 76 m asl.). BARO is
located on the top of a small hill in the urban area of Qingdao, and around 0.7 km away from the coastline of the
Yellow Sea (Figure S1). $PM_{2.5}$ samples were collected separately during daytime and nighttime. Field blank
samples were also collected during the campaign by placing filters in the samplers with the samplers switched off.
After the sampling process, $PM_{2.5}$ samples were sealed and stored at $-20°C$ before analysis.
For the measurement of water-soluble ions (WSIs) and carbonaceous matters, aerosol samples collected on QM-
A filters were utilized. The samples collected on Whatman® 41 filters were used for the detection of elements.
Firstly, samples were cut into pieces and immersed in Milli-Q pure water. Then, water-soluble matters were
extracted by ultrasonic vibration at approximately 0°C for 40 min. The water extracts were then filtered through
syringes with 0.45 μm strainer heads (PALL). The filtered extracts were analyzed for WSIs, including $Na^+$, $NH_4^+$,
$K^+$, $Mg^{2+}$, $Ca^{2+}$, $F^-$, $Cl^-$, $SO_4^{2-}$, $NO_3^-$, $C_2O_4^{2-}$, using ion chromatography (IC, Dionex ICS-3000, Dionex Corp.,
Sunnyvale, CA, USA). Similar sample pretreatment procedures were used to determine soluble elements. While,
10 ml of filtrate was taken and 0.187 ml $HNO_3$ (mass fraction: 69%) was added to water extracts before measuring
soluble elements, in case soluble Fe(II) was oxidized into an insoluble state. To determine total elements, sample
pieces were placed into inner-tanks and subjected to digestion with a mixture of $HNO_3$ + HF (at a volume ratio of
4:1) at 180°C for 48 h. The element concentrations were measured using inductively coupled plasma mass
spectrometry (ICP-MS, Model: iCAP Qc, Thermo Fisher Scientific Inc., Germany). Carbonaceous materials,
specifically organic carbon (OC) and elemental carbon (EC), were analyzed using a sunset OC/EC analyzer from
Sunset Laboratory Inc. The detection limits of the analysis instruments used can be found in Table S1. The organic
matter (OM) content was estimated with 1.6 times OC, as proposed by Turpin and Lim (2001). Further details
about sample collection, pretreatment procedures, and chemical species detection can be found in our previous
work (Li et al., 2023a; Li et al., 2023b).
Various weather conditions and air pollution characteristics were encountered during the observation period,
including clean, slightly-polluted (SP), heavily-polluted, foggy, and dusty conditions. Due to the large deviations
and uncertainties in the statistical results of dust-related samples, data from these samples were not considered.
Additionally, samples from heavily-polluted periods (N = 6, defined by $PM_{2.5} > 50$ μg m$^{-3}$ and $PM_{2.5}/PM_{10} > 0.4$)
and fog-influenced samples (N = 12) were also not included because of the limited sample number and the
significant difference in fog durations between samples. In this paper, we focus on the results of the clean period
samples (N = 19) and the SP period samples (N = 32). Clean periods samples were collected when $PM_{2.5} < 30$ μg
m$^{-3}$ and $PM_{10} < 50$ μg m$^{-3}$. The SP periods samples were those collected when $30$ μg m$^{-3} < PM_{2.5} < 50$ μg m$^{-3}$ and
those collected when $PM_{2.5} < 30$ μg m$^{-3}$ while $PM_{10} > 50$ μg m$^{-3}$.

**2.2 Aerosol pH and liquid water content**

ISORROPIA (version II) thermodynamic equilibrium model was employed to estimate gas concentrations and
aerosol water pH (Song et al., 2018). The forward mode, which uses both gas and aerosol data as model input, was
utilized for pH calculations. This approach was preferred to the reverse mode because the later, using only aerosol
data, is very sensitive to the uncertainties of the measured WSIs concentrations (Hennigan et al., 2015; Song et al.,
2018). "Metastable-mode" was employed in ISORROPIA, assuming that solid precipitates did not form except for
$CaSO_4$. The concentrations of gaseous species (i.e., $NH_3(g)$, $HNO_3(g)$, $HCl(g)$) were not measured at the site. In
alignment with the approach proposed by Sun et al. (2018), we devised a strategy to estimate the concentrations
of these gaseous species. Initially, the input of aerosol data was assumed as the sum of aerosol and gas data
(specifically for $HNO_3$, $HCl$ and $NH_3$). This step provided us with the first set of gas and aerosol data outputs. For
the second run, the gas data output derived from the initial run was added to the original aerosol data, and it was
considered as the sum of gas data and aerosol data just like the first run to calculate $HNO_3(g)$, $HCl(g)$ and $NH_3(g)$.
The same method was employed for subsequent iterations until the variance in the $NO_3^-$ output below the 1%
threshold in mass. The calculation processes can be described by the following equations:

$$Input[C_{Aerosol}+C_{Gas}]_{N+1}=C_{Aerosol}+[C_{Gas}]_N \qquad (1)$$

$$L = \left| \frac{[C_{NO_3^-}]_{N+1} - [C_{NO_3^-}]_N}{[C_{NO_3^-}]_N} \right| \times 100\% \qquad (2)$$

where $C_{Aerosol}$ is the observed concentration of $NO_3^-$ (or $NH_4^+$, $Cl^-$), $C_{Gas}$ is the concentration of gaseous species
of $HNO_3(g)$ (or $NH_3(g)$, $HCl(g)$), and $[C_{Gas}]_N$ is the concentration of gaseous species of $HNO_3(g)$ (or $NH_3(g)$,
$HCl(g)$) output by ISORROPIA in the $N^{th}$ run ($N \geq 1$). The iteration was stopped until $L < 1\%$.
Finally, three times of iterations ($N_{max} = 3$) were determined when $L = 0.1\%$. The aerosol pH was calculated by
using aqueous $H^+$ concentration and aerosol liquid water content (ALWC) outputted by ISORROPIA, as described
by equation (3).

$$pH = -\log_{10} \frac{1000 \times H^+(aq)}{ALWC} \qquad (3)$$

Significant correlations between the results of the first run and the fourth run were observed for pH ($r^2 = 0.95$) and
ALWC ($r^2 = 0.99$), indicating the stability and reliability in estimating the pH and ALWC by ISORROPIA II
(Figure S2). Moreover, the correlations of $NO_3^-$ ($r^2 = 0.71$), $NH_4^+$ ($r^2 = 0.98$) and $Cl^-$ ($r^2 = 0.51$) between the
simulated results and measured concentrations are significant, demonstrating the robust confidence level of the
simulated results (Figure S3).
In addition, the impact of organic matter (OM) on aerosol pH was determined to be minimal. This can be attributed
to the limited sensitivity of the predicted pH to the water uptake by organic species ($ALWC_{org}$) when the OM
fraction in $PM_{2.5}$ is low (Guo et al., 2015; Liu et al., 2017). Following the methods of Guo et al. (2015), we
estimated $ALWC_{org}$ and its influence on aerosol pH. Our analysis determined the $ALWC_{org}$ to range between 0.83
and 3.31 μg m$^{-3}$, constituting merely 2.6−9.8% of the total ALWC. Aerosol pH was about 0.03−0.08 higher when
considering OM, affirming the negligible effect of OM on aerosol pH (see Text S1 in the supporting information
for more details).
**2.3 Weather conditions and air quality data**
The publicly released temperature, RH, surface pressure, wind speed, and wind direction recorded every 10
minutes were obtained from a meteorological observatory of the Qingdao Meteorological Bureau (Figure S1).
Hourly mass concentrations of PM$_{2.5}$, PM$_{10}$, SO$_2$, NO$_2$, O$_3$ and CO were obtained from an adjacent air quality
monitoring station in the Shinan District of Qingdao City (Figure S1), which is managed by Ministry of Ecology
and Environment of the People's Republic of China (http://www.mee.gov.cn/).
To examine the relative abundance of chemical species in aerosols, we reconstructed the mass concentrations of
PM$_{2.5}$ by equation (4) using the obtained concentrations of WSIs, OM, EC and elements.
$$PM_{2.5R} = WSIs + OM + EC + Elements + Si + Ca \tag{4}$$
where PM$_{2.5R}$ is the reconstructed PM$_{2.5}$, and WSIs consists of Na$^+$, NH$_4^+$, K$^+$, F$^-$, Cl$^-$, SO$_4^{2-}$, NO$_3^-$ and C$_2$O$_4^{2-}$. As
for elements, Mg, Al, V, Cr, Mn, Fe, Ni, Co, Cu, Zn, Ga, As, Se, Rb, Sr, Cd, Ba, Tl and Pb were considered. Si
and Ca concentrations were estimated based on the mass ratio of Si/Al (3.43) following the methodology described
by Huang et al. (2010) and the mass ratio of Ca/Al (0.80) suggested by Arimoto et al. (2004) and Wang et al.
(2011). Because the nearby monitoring station is closer to the sea and less affected by human activities (yellow
dot in Figure S1), the level of PM$_{2.5R}$ is higher than the observations from the monitoring station. But the trends of
variations of these two datasets were consistent, indicating the high confidence of the PM$_{2.5R}$ dataset. In addition,
any mention of ionic ratios or normalized parameters in the results and discussions of this paper indicates the data
was divided by PM$_{2.5R}$.
**2.4 Provenances of air masses**
The HYbrid Single-particle Lagrangian Integrated Trajectory (HYSPLIT) Model
(https://www.ready.noaa.gov/HYSPLIT.php), developed by NOAA, was applied to calculate the origins of air
masses from which PM$_{2.5}$ samples were collected. Gridded GDAS data with a horizontal resolution of 1.0° × 1.0°
were used as the input. Backward trajectories were computed for a period of 48 h, with starting points located at
300 m above ground level.

## 3 Results

### 3.1 Meteorological features of clean and SP periods

During clean periods, the backward trajectories reveal that the air masses mainly originated from sea areas (Figure 1). The prevailing sea breeze resulted in high RH levels of 81.5 ± 4.9% during daytime and 86.6 ± 8.8% during nighttime (Table 1). The minimal temperature variance of less than 2℃ between daytime and nighttime further reflects the characteristics of marine atmosphere. In contrast, air masses during SP periods originated from various directions, with a significant number traversing terrestrial regions prior to arriving at the collection site. Temperature and RH exhibited noticeable diurnal variations. The daytime temperature was 17.2 ± 3.0℃ and decreased to 13.2 ± 3.7℃ during nighttime. The RH levels were 62.1 ± 9.4% and 76.8 ± 9.4% during daytime and nighttime, respectively.

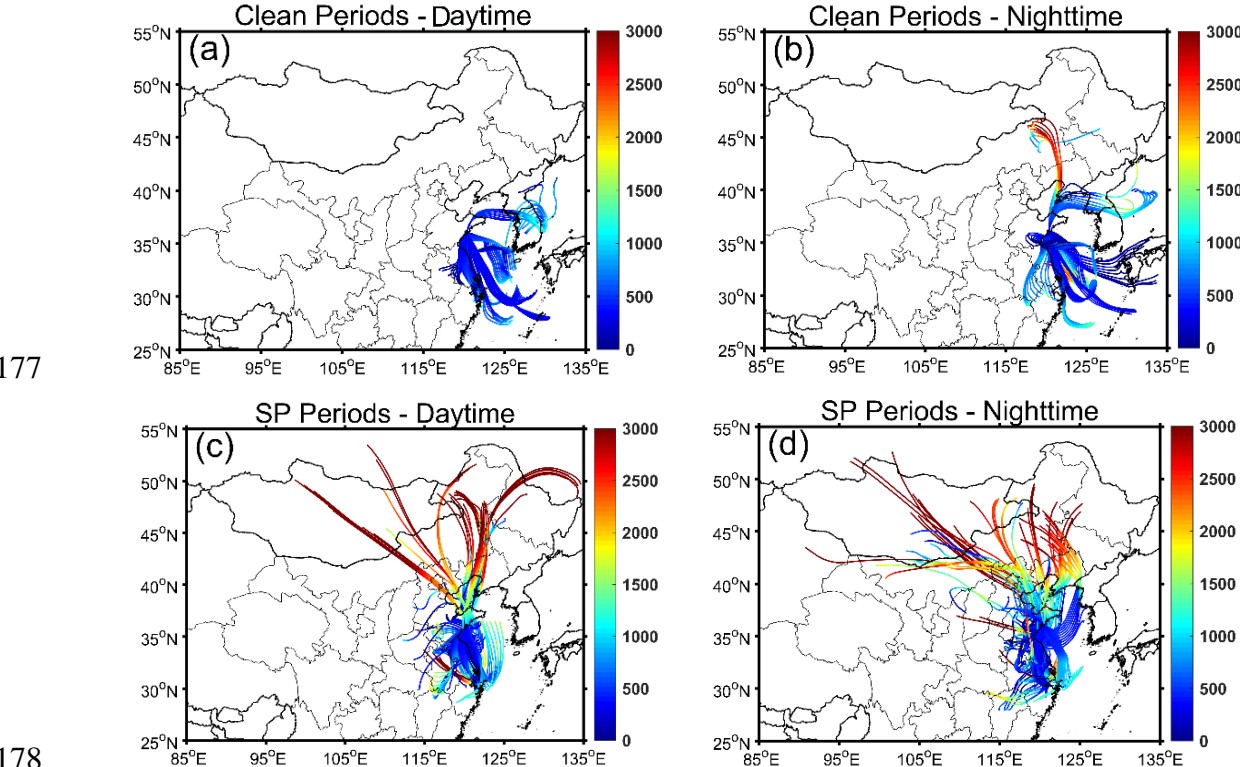

**Figure 1: 48-h backward trajectories during daytime and nighttime during clean and slightly-polluted (SP) periods. Trajectories are color-coded based on the altitude (unit: m) above the ground.**

### 3.2 Concentrations of PM$_{2.5}$ and Fe, and %Fe$_S$

Table 1 presents the PM$_{2.5}$ levels and aerosol Fe concentrations during both clean and SP periods. Under clean conditions, PM$_{2.5}$ concentrations were similar during daytime and nighttime, with average values of 16.9 μg m$^{-3}$ and 16.4 μg m$^{-3}$, respectively. Compared to the nighttime, Fe$_T$ and Fe$_S$ concentrations were higher during the

daytime, which were $289.2 \pm 223.4$ ng m$^{-3}$ and $20.0 \pm 10.5$ ng m$^{-3}$, respectively. Daytime levels of Fe$_T$ and Fe$_S$ were 1.5 times and 1.6 times as high as those observed at night, respectively. The increase in Fe$_T$ and Fe$_S$ during daytime may be linked to heightened human activities. Furthermore, the elevated Fe$_S$ during daytime could be attributed to photochemical processes, which promoted the dissolution of aerosol Fe, a topic to be discussed further in Section 4.2. %Fe$_S$ values ranged from 2.3% to 14.1% with an average of 8.7% during daytime, approximately 1.4 times the nighttime average of 6.3% (after removing an extreme point of 37.2%).

**Table 1. Meteorological parameters, %Fe$_S$, aerosol pH, the concentrations (average ± standard deviation) of PM$_{2.5}$ and chemical species during clean and slightly-polluted periods.**

| | Clean Periods | | Slightly-polluted Periods | |
|---|---|---|---|---|
| | Daytime | Nighttime | Daytime | Nighttime |
| PM$_{2.5}$ (μg m$^{-3}$) | $16.9 \pm 3.1$ | $16.4 \pm 5.6$ | $30.3 \pm 7.0$ | $28.3 \pm 7.7$ |
| Temperature (°C) | $16.6 \pm 2.8$ | $14.3 \pm 2.3$ | $17.2 \pm 3.0$ | $13.2 \pm 3.7$ |
| RH (%) | $81.5 \pm 4.9$ | $86.6 \pm 8.8$ | $62.1 \pm 9.4$ | $76.8 \pm 9.4$ |
| ALWC (μg m$^{-3}$) | $30.0 \pm 13.5$ | $55.0 \pm 53.7$ | $22.5 \pm 13.2$ | $44.1 \pm 33.8$ |
| Fe$_T$ (ng m$^{-3}$) | $289.2 \pm 223.4$ | $186.7 \pm 122.2$ | $938.3 \pm 850.5$ | $520.3 \pm 496.1$ |
| Fe$_S$ (ng m$^{-3}$) | $20.0 \pm 10.5$ | $12.5 \pm 7.4$ | $25.7 \pm 10.5$ | $21.6 \pm 8.1$ |
| %Fe$_S$ (%) | $8.7 \pm 3.8$ | $6.3 \pm 4.1$ | $3.7 \pm 2.0$ | $5.8 \pm 3.0$ |
| pH | $0.46 \pm 0.83$ | $1.06 \pm 0.96$ | $1.16 \pm 0.88$ | $0.98 \pm 0.75$ |
| SO$_4^{2-}$ (μg m$^{-3}$) | $13.97 \pm 5.19$ | $10.97 \pm 8.06$ | $14.94 \pm 5.81$ | $13.78 \pm 5.43$ |
| F(SO$_4^{2-}$)[a] | $42.9\% \pm 14.0\%$ | $36.8\% \pm 14.0\%$ | $20.9\% \pm 3.6\%$ | $23.0\% \pm 5.3\%$ |
| NO$_3^-$ (μg m$^{-3}$) | $5.82 \pm 3.49$ | $5.63 \pm 4.87$ | $26.71 \pm 13.15$ | $22.80 \pm 10.81$ |
| F(NO$_3^-$)[b] | $15.7\% \pm 6.0\%$ | $17.7\% \pm 11.3\%$ | $35.4\% \pm 9.0\%$ | $35.6\% \pm 9.0\%$ |
| (2[SO$_4^{2-}$]+[NO$_3^-$])/PM$_{2.5R}$ (μmol μg$^{-1}$) | $0.0115 \pm 0.0026$ | $0.0105 \pm 0.0023$ | $0.0101 \pm 0.0017$ | $0.0106 \pm 0.0016$ |

[a] F(SO$_4^{2-}$) is the fraction of SO$_4^{2-}$ in PM$_{2.5}$ mass, which was calculated by using SO$_4^{2-}$ concentrations divided by PM$_{2.5R}$ concentrations. [b] F(NO$_3^-$) is the fraction of NO$_3^-$ in PM$_{2.5}$ mass. The calculation method is the same as F(SO$_4^{2-}$).

Under SP conditions, PM$_{2.5}$ was at similar levels during daytime and nighttime with the average values of 30.3 μg m$^{-3}$ and 28.3 μg m$^{-3}$, respectively. However, the daytime Fe$_T$ ($938.3 \pm 850.5$ ng m$^{-3}$) was much higher than the nighttime Fe$_T$ ($520.3 \pm 496.1$ ng m$^{-3}$), which was approximately threefold higher than during clean periods. Similarly, the daytime Fe$_S$ concentration of $25.7 \pm 10.5$ ng m$^{-3}$ was also slightly higher than the nighttime concentrations of $21.6 \pm 8.1$ ng m$^{-3}$, which was 1–2 times higher than that during the clean period. Different from

the clean period, %Fe$_S$ was markedly higher at night (5.8% ± 3.0%) compared to the daytime %Fe$_S$ (3.7% ± 2.0%)
during the SP period, ranging from 1.0% to 12.3%.

**3.3 Chemical characteristics of PM$_{2.5}$**

Figure 2 illustrates the mass fractions of various chemical species present in the reconstructed PM$_{2.5}$ (PM$_{2.5R}$).
During the clean period, WSIs were the dominant components, constituting about 75.0% and 74.1% of PM$_{2.5}$ mass
during daytime and nighttime, respectively. SO$_4^{2-}$, NO$_3^-$, and NH$_4^+$ were the main contributors to WSIs. During
daytime, SO$_4^{2-}$ and NO$_3^-$ were 13.97 ± 5.19 μg m$^{-3}$ and 5.82 ± 3.49 μg m$^{-3}$, respectively, serving as the major
acidic species and accounting for 42.9% and 15.7% of the PM$_{2.5}$ mass (Table 1 and Figure 2). At night, SO$_4^{2-}$ and
NO$_3^-$ concentrations decreased slightly, which were 10.97 ± 8.06 μg m$^{-3}$ and 5.63 ± 4.87 μg m$^{-3}$, respectively,
representing 36.8% and 17.7% of the PM$_{2.5}$ mass (Table 1 and Figure 2). In other words, the two main acid species,
SO$_4^{2-}$ and NO$_3^-$, occupied slightly larger fractions of the PM$_{2.5}$ mass during the daytime (58.7%) compared to the
nighttime (54.6%), along with the lower ALWC, resulting in the lower aerosol pH of 0.46 ± 0.83 during daytime
(Table 1). At night, aerosol pH (1.06 ± 0.96) increased by a factor of 2.3 compared to daytime.
The aerosol pH calculated in this work was evidently lower than many other areas of China (Liu et al., 2017; Wang
et al., 2019; Xu et al., 2020). During the clean period, air masses mainly originated from the seas. Therefore, the
aerosol pH can be very acidic because of the lack of sources of alkaline substances over the ocean, such as NH$_3$,
Ca$^{2+}$, et al. (Zhou et al., 2018). Compared to the inland areas, much lower aerosol pH in coastal areas is reasonable
(Wang et al., 2022). For instance, Zhou et al. (2018) reported that the pH of aerosols near the Bohai Sea can be as
low as around 1.0. Moreover, they also found that the daytime aerosol acidity was significantly stronger than that
during the nighttime in coastal areas. This observation aligns with the findings during clean periods in our study,
which were characterized by the predominance of sea breezes. In this study, we employed the ratio of acidic
substances to PM, namely, (2[SO$_4^{2-}$] + [NO$_3^-$])/PM$_{2.5R}$, to characterize the level of acidic substances in a unit of
PM$_{2.5}$, because SO$_4^{2-}$ and NO$_3^-$ were predominant acidic species within WSIs (>75% in mass). It was 0.0115 ±
0.0026 μmol μg$^{-1}$ and 0.0105 ± 0.0023 μmol μg$^{-1}$ in PM$_{2.5}$ mass during daytime and nighttime, respectively (Table

226      1).

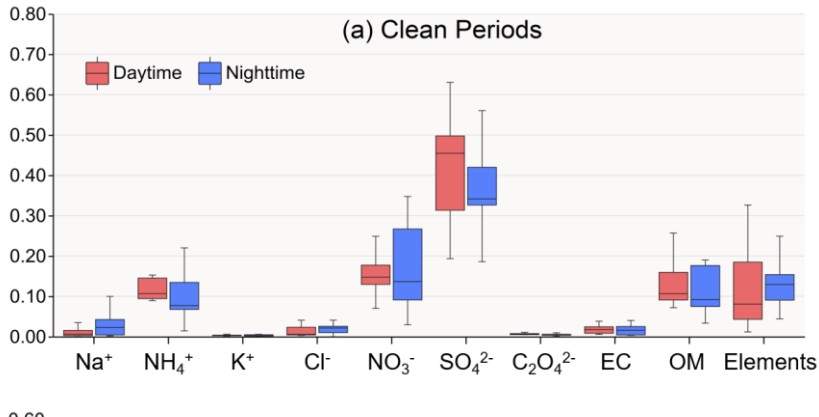
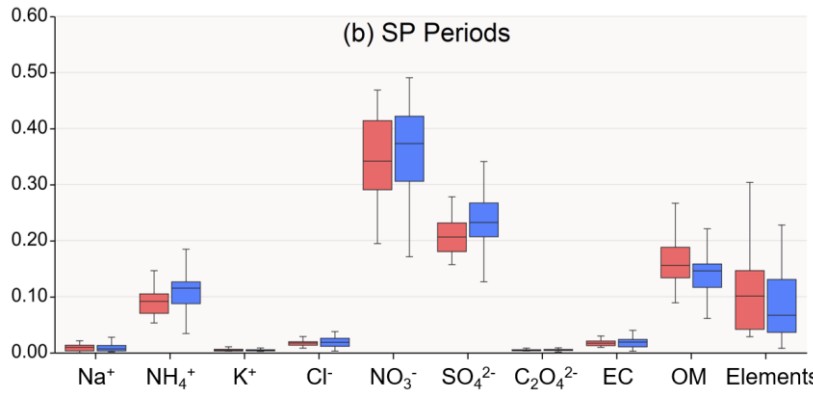
**Figure 2: Mass fractions of chemical species in reconstructed PM₂.₅ mass during daytime and nighttime in clean and SP**
**conditions. Mg²⁺ and Ca²⁺ are not shown in the above pictures, because total Mg is included in elements data and total**
**Ca is assessed by 0.8 times Al.**

During the SP period, WSIs retained similar proportions in $PM_{2.5}$ as during the clean period, accounting for 70.5%
and 74.3% during daytime and nighttime, respectively. $SO_4^{2-}$, $NO_3^-$, and $NH_4^+$ were also the main contributors to
WSIs. In the daytime, the concentrations of $SO_4^{2-}$ and $NO_3^-$ were 14.94 ± 5.81 μg m⁻³ and 26.71 ± 13.15 μg m⁻³,
respectively, showing a marginal elevation over nighttime levels (Table 1). However, $SO_4^{2-}$ had evidently lower
contributions to $PM_{2.5}$ compared to the clean period, which were only 20.9% and 23.0% during daytime and
nighttime, respectively (Table 1 and Figure 2). In contrast, $NO_3^-$ had a noticeably higher contribution to $PM_{2.5}$
compared to the clean period, exhibiting little diurnal variation, with percentages of 35.4% and 35.6% during
daytime and nighttime, respectively (Table 1 and Figure 2). In total, the ratio of acids to PM (i.e.,
$(2[SO_4^{2-}]+[NO_3^-])/PM_{2.5R})$ was 0.0101 ± 0.0017 μmol μg⁻¹ during daytime and 0.0106 ± 0.0016 μmol μg⁻¹ during
nighttime (Table 1). Even though the ALWC (44.1 ± 33.8 μg m⁻³) was significantly more abundant at night
compared to the daytime (22.5 ± 13.2 μg m⁻³), the aerosol pH was lower at night. Specifically, the nighttime
aerosol pH was 0.98 ± 0.75, while the daytime aerosol pH was slightly higher at 1.16 ± 0.88, indicating weaker
aerosol acidity during daytime with a 18.4% increase in pH compared to nighttime aerosols.

## 4 Discussion

We found that daytime %Fe$_S$ was much higher than nighttime %Fe$_S$ during the clean period, while the opposite pattern emerged during the SP period. This section delves into the primary factors driving the distinct diurnal shifts in aerosol %Fe$_S$ during clean and SP periods, based on the aspects of aqueous-phase conversions and photocatalysis reactions.

### 4.1 Aqueous-phase conversions promoted by acid processes

The %Fe$_S$ was dependent on the acidification of the aerosol particles, and high %Fe$_S$ was associated with low aerosol pH (Table 1). The pH of aerosols is controlled by ALWC and H$^+$ contents. The predominant acidic species, i.e., SO$_4^{2-}$ and NO$_3^-$, play crucial roles in promoting the dissolution of insoluble Fe through proton-promoted reactions. As shown in Figure 3a, there was a significant negative correlation between the aerosol pH and the relative content of these two acidic species when the pH was below 4. Especially during clean and SP periods (r = 0.62, Figure 3a), the slope of the regression line was approximately –602.99, indicating that a variation of 1.0 nmol μg$^{-1}$ of the acidic species content in PM$_{2.5}$ can lead to a noticeable fluctuation of aerosol pH (about 0.60). For instance, the daytime aerosol pH was 0.60 lower than that of the nighttime during the clean period, even though the difference of the two acidic species content was only about 1.0 nmol μg$^{-1}$.

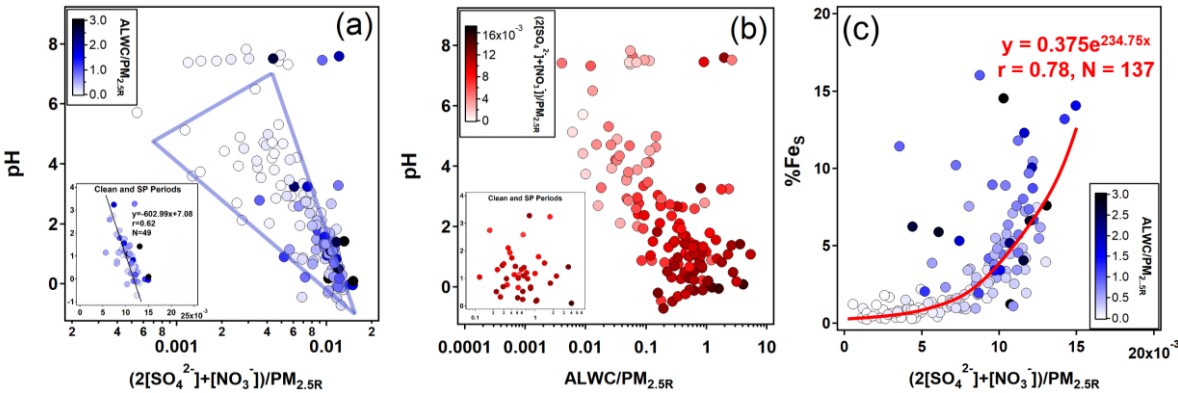

**Figure 3: Relationships among pH, the normalized relative abundance of ALWC (unit: μg m$^{-3}$) and main acidic species (= 2[SO$_4^{2-}$] + [NO$_3^-$], unit: μmol m$^{-3}$) with respect to the reconstructed PM$_{2.5}$ (i.e., PM$_{2.5R}$, unit: μg m$^{-3}$), and %Fe$_S$. The subgraph at the bottom-left of figures (a) and (b) show scatter plots during clean and SP periods with the linear regression line obtained by using the Igor Pro-based program developed by Wu and Yu (2018).**

There was no prominent correlation between pH and ALWC when the pH exceeded 6 (Figure 3b). When the pH was smaller than 6, the increasing ALWC facilitated the heterogeneous reactions of SO$_2$ and NO$_2$ to generate more

$SO_4^{2-}$ and $NO_3^-$, lowering the aerosol pH and enhancing the %Fe$_S$. The formation of $SO_4^{2-}$ and $NO_3^-$ will further
facilitate the growth of ALWC due to their remarkable hygroscopicity, establishing a positive feedback (Path A in
Figure 4), referred to as the "ALWC-acid" feedback (Wang et al., 2016; Wu et al., 2018b). On the other hand,
ALWC dilutes $H^+$ in aerosol water. This process weakens the aerosol acidity and inhibits the particles from %Fe$_S$
elevation (Path B in Figure 4). In addition, the increasing ALWC served as a medium for loading water-soluble
components may promote the formation of Fe$_S$ (Path C in Figure 4). The profound influence of acidic species on
the aerosol pH indicates the predominance of the "ALWC-acid" feedback in modulating the aerosol pH and
augmenting %Fe$_S$ (Figures 3a, 3c and S4). The high %Fe$_S$ we observed during daytime and nighttime can be
attributed to the relatively higher content of acidic species in PM$_{2.5}$.

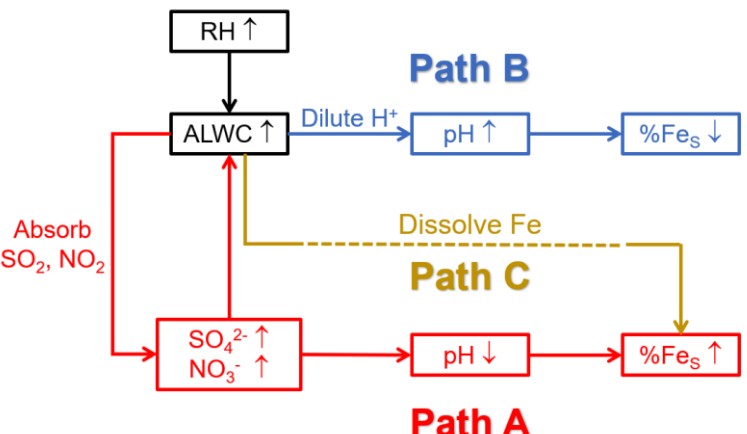

**Figure 4: Schematic diagram of ALWC affecting pH and %Fe$_S$. Path C is challenging to observe and quantify because**
**of the Fe$_S$ extraction using MilliQ water in the sample pretreatment.**

RH is a key factor in the formation of $SO_4^{2-}$ and $NO_3^-$ through heterogeneous/aqueous-phase reactions within
aerosols (Wang et al., 2016; Liu et al., 2020; Hou et al., 2022). As demonstrated in Figure 5, the strong dependency
of the oxidation rate of sulfur (SOR, defined as $[SO_4^{2-}]/([SO_4^{2-}] + [SO_2])$) on RH was observed under moderate
humid conditions (r = 0.64, p < 0.01). But the nitrogen (NOR, defined as $[NO_3^-]/([NO_3^-] + [NO_2])$) had a poor
dependence on RH (r = 0.46, p > 0.05). A decrease of 10% in RH resulted in a notable reduction of 7.6% in SOR
(Figure 5). Such a striking RH dependence was observed mainly during the SP period, indicating the significant
role of heterogeneous reactions in controlling the formation of $SO_4^{2-}$. Therefore, the facilitation of aqueous-phase
conversions leading to the formation of $SO_4^{2-}$ was more pronounced at night during the SP period, attributed to
the high RH. This, in turn, resulted in a high proportion of $SO_4^{2-}$ and acidic species, as well as the elevated SOR
(Table 1, Figures 2b and S5). The nighttime aerosol pH was approximately 0.18 units lower than that during
daytime, but this slight variation did not hinder the efficient formation of $Fe_S$ during nighttime in SP periods.
In contrast, RH was generally above 80% during daytime and nighttime in clean periods. The SOR was 0.49 on
average and did not exhibit a clear correlation with RH beyond 78% (Figure 5a). Similar phenomena have been
observed in previous studies, suggesting the existence of a saturation point in the promotion of RH on the aqueous-
phase formation of $SO_4^{2-}$ (Wang et al., 2019; Wang et al., 2021). High RH ($> 70\%$) can cause water-soluble species
to deliquesce and form an aqueous layer on the particle surface. Once the aqueous layer forms, the influence of
RH variations becomes minimal (Shi et al., 2022). Hence, the degree of aqueous-phase processes promoting $SO_4^{2-}$
formation during clean periods was similar across both daytime and nighttime.

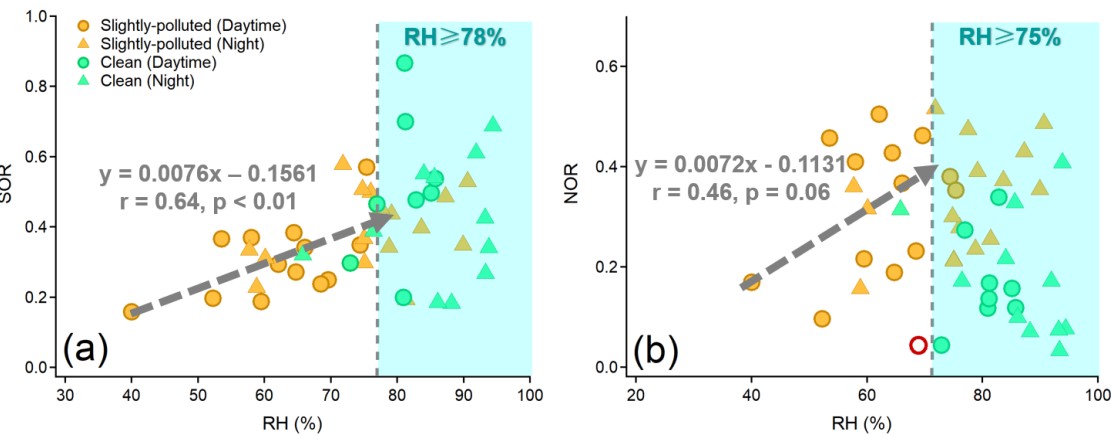


**Figure 5: The dependence of SOR (a) and NOR (b) on RH during clean and slightly-polluted periods. The fitting of the**
**regression line between SOR and RH was fitted when RH<78%. The fitting of the regression line between NOR and RH**
**was fitted when RH<75% and one deviation point (the red circle in (b)) was removed.**

**4.2 Daytime enhancement by photocatalysis reactions**
**4.2.1 The influence of photochemical processes on sulfate formation**
Photochemical reactions can enhance the formation of acidic species and increase the aerosol %$Fe_S$ through aerosol
acidification (Tao et al., 2020; Liu et al., 2021a). The large proportion of acidic species during the daytime of the
clean period was attributable to $SO_4^{2-}$, which was 6.1% higher than the nighttime $SO_4^{2-}$ (Table 1 and Figure 2a).
Despite similar levels of $SO_2$ observed during daytime and nighttime, the daytime SOR reached as high as $0.50 \pm$
0.20 (Figure S5). The conversion rates in the aqueous phase were similar during daytime and nighttime in clean
periods. Therefore, the substantial fraction of $SO_4^{2-}$ was most likely caused by photochemical reactions.
$O_x$ (described by the sum of $O_3$ and $NO_2$) was investigated to quantify the potential of photochemical reactions,
following the method of Wu et al. (2018a). The daytime $O_x$ concentration ($56.1 \pm 6.4$ ppb) was about 5.1% higher
than that of nighttime $O_x$ ($53.4 \pm 9.3$ ppb) during the clean period. The substantial SOR occurred under the extreme
$O_x$ conditions (Figure 6a), suggesting a significant contribution of the photochemical reactions during the clean
period. The enhancement of daytime photochemistry and aqueous chemistry on aerosol %Fe$_S$ was more
pronounced than that of the nighttime aqueous reactions solely during the clean period (Figure 7a and 7b).

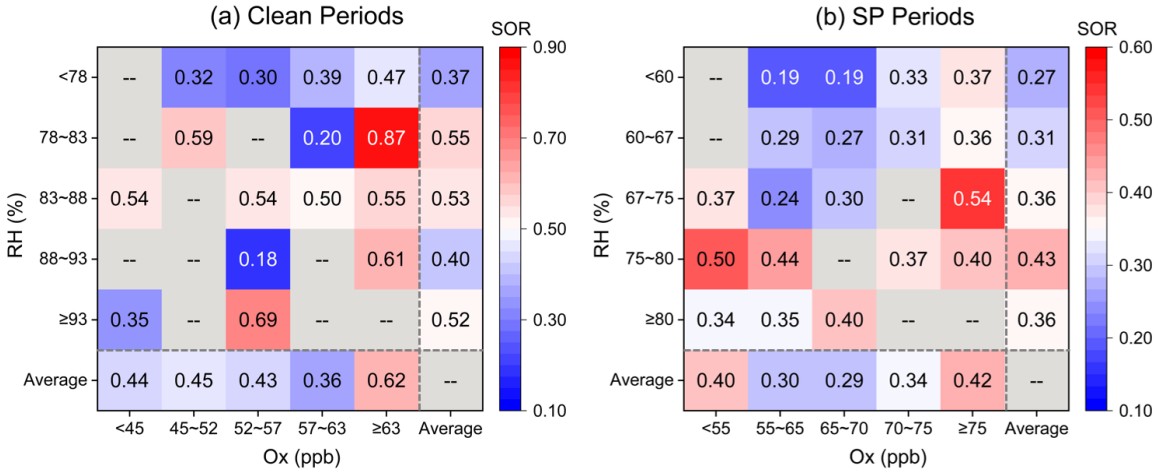


**Figure 6: RH-O$_X$ image plots colored by SOR during clean and SP periods. The last row and last column of the matrices**
**represent the average value of SOR in the corresponding ranges of RH and O$_X$.**

During the SP period, the extent of SOR was more influenced by RH than by $O_x$, especially when RH was below
80% (Figure 6b). Nighttime SOR ($0.37 \pm 0.12$) was approximately 1.2 times higher than the daytime SOR ($0.31 \pm$
$0.11$) even though the daytime $O_x$ was higher than that during nighttime (Figure S5), indicating a greater
contribution of liquid/heterogeneous reactions to the $SO_4^{2-}$ formation than photocatalytic reactions. Similar
findings were reported by Hou et al. (2022), who highlighted the dominant role of humidity rather than $O_x$ in $SO_4^{2-}$
formation in haze intensification. The nighttime exhibited a more significant "ALWC-acid" feedback compared to
the daytime during the SP period. The influence of daytime photochemistry combined with aqueous-phase
reactions was comparatively weaker than nighttime aqueous chemistry, leading to the higher %Fe$_S$ at night (Figure
7c and 7d). Notably, $O_x$ concentrations were significantly higher during the SP period in comparison to the clean
period (Figure S5), indicating more active daytime photocatalytic reactions. However, the impact of aqueous-
phase conversions during the SP nighttime period was relatively weak compared to the nighttime of the clean
period. These results suggest that the role of photocatalytic reactions in $SO_4^{2-}$ formation, and subsequently in the
elevation of aerosol %Fe$_S$, was feeble compared to aqueous-phase conversions.

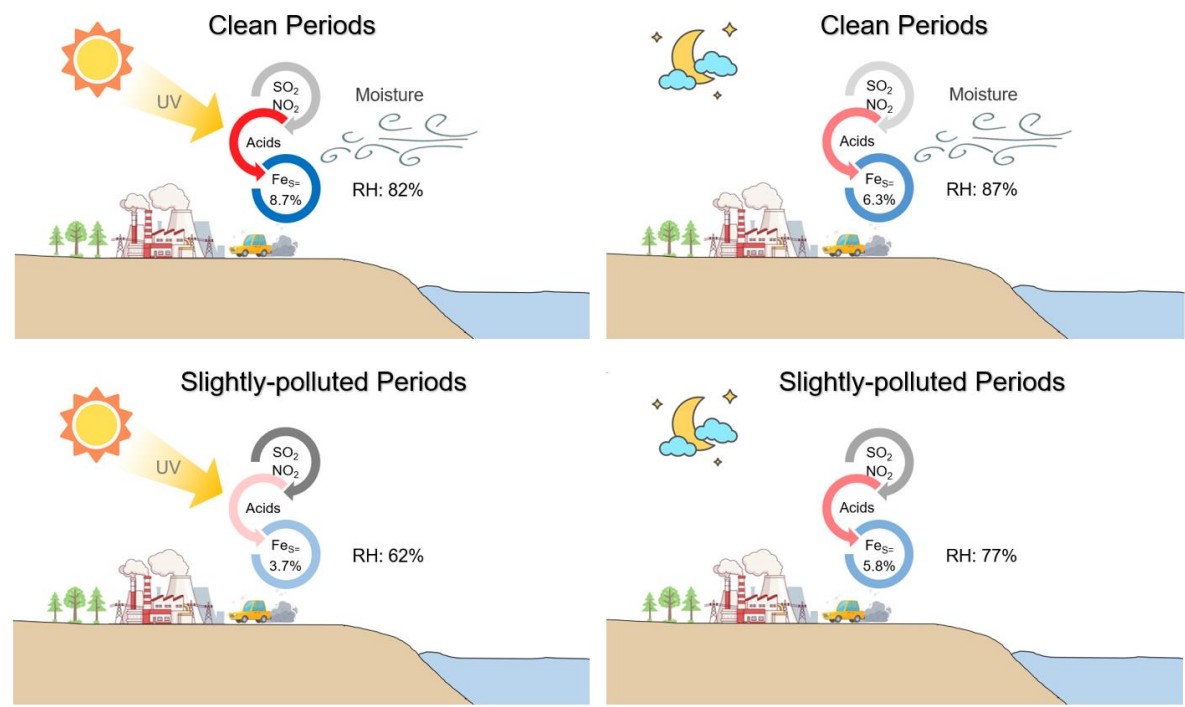

**Figure 7: Conceptual diagram showing the Fe dissolution influenced by acid processes at the coastal city during daytime**
**and nighttime in clean and SP periods.**
Enhancing aerosol %Fe$_S$ through direct photocatalysis pathways is indeed possible. Iron oxides in minerals can
generate conduction band electrons upon irradiation, causing the reductive dissolution of Fe(III)-containing solid
phases to Fe(II) species (Zhang et al., 1993; Fu et al., 2010). However, structural Fe(III), which is the major iron-
related mineral in dust and coal fly ash, does not readily undergo direct reduction upon UV irradiation (Fu et al.,
2012; Fu et al., 2010; Xie et al., 2020). Another pathway for photolysis-conduced iron dissolution involves the
reduction by reactive oxygen species (ROS, e.g., $O_2^{\bullet-}$, $HO_2^{\bullet}$, and $H_2O_2$). These ROS can be generated from
dissolved oxygen accompanied by conduction band electrons, enhancing the Fe dissolution by reducing the solid-
phase Fe(III) into the more soluble Fe(II) form (Zhu et al., 1997; Hettiarachchi and Rubasinghege, 2020). Aerosol
water is necessary for the above reactions, and the proton-promoted dissolution by acid species is indispensable to
dissolve the solid-phase Fe(II) into aerosol solutions. We suppose that the observed weak influence of
photocatalysis on %Fe$_S$ was because of the extreme aerosol acidity. The acidity of aerosols, such as a pH as low
as 2.0 during daytime of the present study, can suppress the contribution of photochemical catalysis in the

formation of $Fe_S$ (Zhu et al., 1993; Fu et al., 2010; Fu et al., 2012). In addition, studies have suggested that Fe

dissolution can be inhibited in $H_2SO_4$ systems under irradiation compared to dark conditions, which could be

another reason for the low $\%Fe_S$ during daytime although the exact mechanism remains unclear (Fu et al., 2010;

Hettiarachchi et al., 2018).

**4.2.2 The enhancement of $\%Fe_S$ promoted by oxalate-related conversions**

Oxalate can form complexes with Fe(III) and participate in photochemical reactions through photoinduced charge

transfer. Oxalate transfers its charge to the Fe(III) surface via photolytic reactions during daytime, resulting in the

reduction of Fe(III) to Fe(II), followed by the dissociation of the formed Fe(II) from the surface and hence the

dissolution of aerosol Fe (Zuo and Hoigne, 1992; Zhang et al., 2019; Lueder et al., 2020). Shi et al. (2022)

identified the oxalate/$Fe_T$ ratio as an excellent predictor for aerosol $\%Fe_S$ through machine learning, underscoring

its remarkable effectiveness. However, field observations rarely confirm its influence on $Fe_S$ from the perspective

of oxalate-Fe photochemistry.

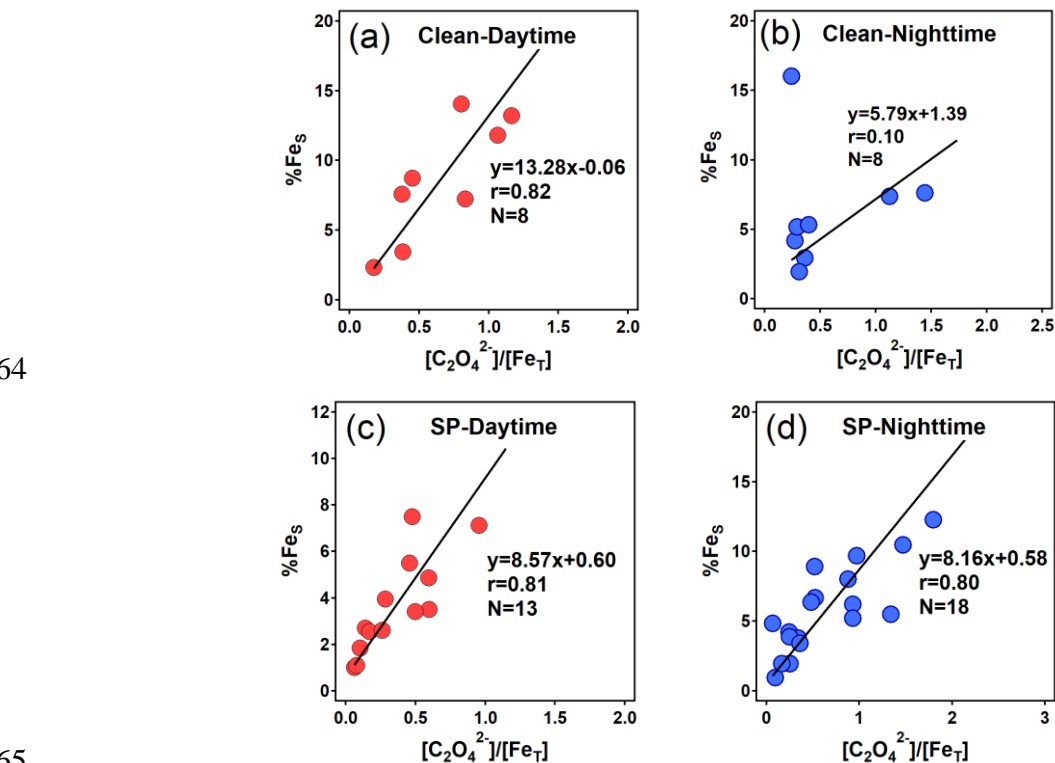

**Figure 8: Relationships between $\%Fe_S$ and the molar ratio (unit: $\mu mol\ \mu mol^{-1}$) of oxalate to $Fe_T$ during daytime and nighttime in clean and SP periods. An extreme point (marked by a pink triangle, $\%Fe_S = 37.2\%$) in (b) was removed.**

In this study, significant correlations were observed between %$Fe_S$ and the molar ratio of [oxalate]/[$Fe_T$] during
daytime in both clean periods (r = 0.82) and SP periods (r = 0.81) (Figure 8a and 8c). Similarly, a striking
correlation was also found in nighttime during the SP period (r = 0.80), although with a lower slope of 8.16 in the
regression line (Figure 8d). Noteworthy is the strong dependence of %$Fe_S$ (or $Fe_S$) on oxalate concentration at
night (Figures 8d, S6b and S6d). Field observations highlight the pivotal role of organic compound complexation
in stabilizing Fe (Sakata et al., 2022). Additionally, as illustrated by Figure 8a and 8c, the variation in %$Fe_S$ induced
by each unit variation in daytime [oxalate]/[$Fe_T$] was greater than its nighttime equivalent. The most notable
increase was observed during the clean period, with a daytime slope of 13.28, marking a 1.6-fold increase over the
SP period (daytime slope = 8.57). Similarly, the concentration of Fes per unit of oxalate showed a parallel trend,
marking the highest daytime slope of the clean period during the campaign (Figure S6). Such patterns imply that
enhanced sunlight in clean days may have catalyzed photochemical processes involving daytime oxalate-Fe,
leading to elevated concentrations of both Fes and %$Fe_S$. While these outcomes have only been discussed through
laboratory simulations (Chen and Grassian, 2013), or indirectly by examining oxalate degradation or sulfate
formation (Zhou et al. 2020), and they have been empirically discovered through field observations now in this
study.
Simultaneously, Fes species redox reactions can facilitate the formation of oxalate in return if the precursors are
abundant, particularly with aqueous-phase reactions playing a pivotal role when RH exceeds 60% (Zhang et al.,
2019). This may elucidate one of the main reasons behind the significant correlations observed between Fes and
oxalate. Notably, oxalate concentration was higher during the daytime compared to the nighttime in this study
(Figure S5), concomitant with elevated Fes concentrations. The photocatalytic degradation of oxalate-Fe,
promoting Fe dissolution during daytime, was unlikely to be the predominant pathway influencing the oxalate
concentration, otherwise a decrease in oxalate concentration would occur (Dou et al., 2021). Therefore, the oxalate
formation process catalyzed by Fes could yield a higher production rate of oxalate during the daytime than at night.
Figure S7 portrays the conceptual diagram of these conversion processes. Similar scenarios might unfold for $SO_4^{2-}$
formation due to the heightened Fe redox reactions during daytime (Zhou et al., 2020). Owing to the extremely
low aerosol pH (< 2), transition-metal ions (TMIs, e.g., Fes)-catalyzed pathway could primarily influence the
secondary formation of $SO_4^{2-}$, leading to potent aerosol acidity (Liu et al., 2021b). The elevated aerosol acidity, in
turn, fostered the formation of Fes, thus furthering the generation of $SO_4^{2-}$ and oxalate under high RH conditions.
The resulting oxalate could then be complexed with Fes, sustaining %$Fe_S$ at a high level at night.
To summarise, the findings of this study suggest that daytime photochemical processes indeed facilitated the
dissolution of aerosol Fe, consequently elevating %Fe$_S$ during the clean period. This mechanism, in turn, may
foster the secondary formation of oxalate and $SO_4^{2-}$. The complexation of organic compounds significantly
contributed to maintaining the high %Fe$_S$ at night. While during SP periods, the diurnal variation in aerosol %Fe$_S$
mainly resulted from the differing levels of aerosol acidity between daytime and nighttime, a conclusion strongly
supported by the higher %Fe$_S$ observed at night compared to daytime.
**4.3 Environmental implications**
Limited research has explored the diurnal variation of aerosol %Fe$_S$. Only an early case investigated the diel
variability of Fe species at an island located in the Caribbean Sea and highlighted the photochemical processing
of Fe (Zhu et al., 1997). This study found a pronounced correlation between Fe$_S$ and acid species within an aerosol
pH range of 0 to 1, emphasizing the considerable influence of aerosol acidification on Fe dissolution. These
findings align with the results of our study. Our results suggest that acid-driven aqueous-phase transformations
could have a more crucial role in altering aerosol %Fe$_S$ than photochemical reactions under certain conditions in
coastal urban areas.
Previous studies pinpointed robust %Fe$_S$ of anthropogenic aerosols, especially for combustion-related fly ash
(Oakes et al., 2012; Wang et al., 2015; Baldo et al., 2022; Li et al., 2022). Unlike urban air, RH tends to be
considerably higher over open oceans, fostering an environment where heterogeneous reactions and the secondary
formation of $SO_4^{2-}$ and $NO_3^-$ are prevalent. In such cases, photochemical reactions and precursors' concentrations
will determine the formation of salts. Given that the air mass of the clean period comprised a mix of marine and
local urban air, it is expected that the Fe dissolution in aerosol particles is an effective way to produce Fe$_S$ during
daylight hours. Air masses moving from densely populated land areas carry substantial amounts of $SO_2$, $NO_2$ and
$NH_3$ to offshore areas, aiding in the formation of $SO_4^{2-}$ and $NO_3^-$ and conducing to the acidic dissolution of Fe in
aerosol particles. Subsequently, the solubilized Fe, through proton-promoted dissolution, can be further stabilized
by the organic complexation of Fe in the marine atmosphere, as indicated by Sakata et al. (2022).
Additionally, the dearth of ammonia sources in the marine atmosphere may hinder the formation of $SO_4^{2-}$ and
$NO_3^-$ to some extent on the one hand (Wang et al., 2017; Guo et al., 2018). The limited availability of ammonia
may be also conducive to enhancing the aerosol acidity and elevating aerosol %Fe$_S$ on the other hand. Considering
that concentrations of HCl in remote marine atmospheric boundary layer are typically higher than in the continent
of East Asia, the influence of chloride on aerosol pH may therefore play a conspicuous role in regulating $\%Fe_S$
(Tobo et al., 2010), on which knowledge is very limited.

**5 Summary**

This study investigated the daytime and nighttime $\%Fe_S$ in $PM_{2.5}$ in a coastal city of China under clean and SP
conditions. Under clean conditions, $\%Fe_S$ was higher during daytime (8.7%) compared to the nighttime (6.3%,
after removing an extreme point of 37.2%). On the contrary, under SP conditions, $\%Fe_S$ was higher at night (5.8%)
than during daytime (3.7%). Significant correlations were observed between the main acidic components ($SO_4^{2-}$
and $NO_3^-$), aerosol pH, and $\%Fe_S$, indicating that the acid process played a dominant role in influencing
aerosol $\%Fe_S$.
The RH consistently exceeded 80% during both daytime and nighttime in clean periods. Aqueous-phase reactions
were found to be most effective in promoting the secondary formation of acid species, with photochemical
processes further enhancing $SO_4^{2-}$ formation during daytime. Together with the lower ALWC, the aerosol pH was
much lower during daytime (0.46 ± 0.83) compared to nighttime (1.06 ± 0.96) during the clean period, which
exerted a more significant influence on aerosol Fe dissolution. In contrast, RH was much higher at night (76.8%)
than that during daytime (62.1%) in the SP period. The dry conditions during daytime notably restricted the
secondary formation of $SO_4^{2-}$ and $NO_3^-$. The acid content in $PM_{2.5}$ was much higher at night under the promotion
of heterogeneous processes, resulting in stronger aerosol acidity and higher aerosol $\%Fe_S$. Furthermore,
photochemical reactions associated with oxalate likely played a considerable role in enhancing $\%Fe_S$ during
daylight hours, a trend more noticeable during the clean period. Oxalate might also be crucial in sustaining
elevated $\%Fe_S$ at night during the SP period.
This study provides insights into the mechanisms of aerosol $\%Fe_S$ modulation in the coastal city. The robust
promotion of aqueous-phase processes and the comparatively weaker influence of photochemistry on enhancing
aerosol $\%Fe_S$ were observed. In urban air, RH was a crucial factor in controlling $\%Fe_S$ through modulating the
heterogeneous reactions of $SO_4^{2-}$ and $NO_3^-$. In contrast, in the oceanic atmospheric boundary layer, precursors'
levels and photochemical processes may be the decisive manipulators on aerosol $\%Fe_S$. Therefore, the content of
bioavailable Fe in urban-related aerosols may be greatly elevated after intrusion into the marine atmosphere, which
holds significant importance for future research.

*Author contributions.* WL: investigation, formal analysis, writing – original draft, writing – review and editing; YQ: methodology; YL: methodology; GW: visualization; YZ: methodology; JS: methodology; WQ: methodology; LS: supervision, funding acquisition; WW: methodology; DZ: funding acquisition, methodology, writing – review and editing; YZ: conceptualization, funding acquisition, methodology, supervision, writing – review and editing.

*Acknowledgements.* We gratefully acknowledge the National Oceanic and Atmospheric Administration (NOAA) Air Resources Laboratory (ARL) for the provision of the HYSPLIT transport and dispersion model, available at (https://www.ready.noaa.gov/HYSPLIT.php), and the Global Data Assimilation System (GDAS). Additionally, we acknowledge the use of ISORROPIA II, accessible at (https://www.epfl.ch/labs/lapi/models-and-software/isorropia/), developed by the Schools of Earth & Atmospheric Sciences and Chemical & Biomolecular Engineering at the Georgia Institute of Technology, for the calculation of aerosol pH and liquid water content.

*Competing interests.* The authors declare that they have no conflict of interests.

*Financial support.* This research was supported by National Natural Science Foundation of China (Grant Number: 41875155, 41605114, 41875174) and the Overseas Joint Training Program for Doctoral Students of Ocean University of China. D.Z. was supported by JSPS KAKENHI 21H01158.

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
