# Peer review of "Daytime and nighttime aerosol soluble iron formation in clean 1"

_EGUsphere, 2023_

## Author Comment (AC1)

**Point by Point Response to Review Comments**

**Daytime and nighttime aerosol soluble iron formation in clean and slightly-polluted moisture air in a coastal city in eastern China**
* * *
We thank the **Reviewer #1** for the detailed and constructive comments. We provide below point-by-point response to the comments. The reviewer's comments and the original contents of the manuscript are in **black**. The response text is in **blue**. Revisions in the manuscript are in **red**.

**General comments:**

This manuscript investigated the components of total and soluble iron in $PM_{2.5}$ in both daytime and nighttime in a coastal city, and tried to evaluate the effects of aqueous-phase and photochemical reactions on that. The topic is interesting and the discussion is comprehensive.

However, the language needs to be extensively improved throughout the manuscript, and the content could be more concise. Moreover, the references need to be double-checked. The original source should be cited as much as possible. For example, the original source of ISORROPIA II should be acknowledged as well.

**General response:**

In the revised manuscript, we have enhanced the linguistic expression, rectified grammatical inaccuracies, and rendered the descriptions more succinct. Moreover, we have conducted a thorough verification of the references, ensuring the inclusion of the original literature. Additionally, an acknowledgment has been extended to ISORROPIA II, as detailed below:

"*Acknowledgements. We gratefully acknowledge the National Oceanic and Atmospheric Administration (NOAA) Air Resources Laboratory (ARL) for the provision of the HYSPLIT transport and dispersion model, available at (https://www.ready.noaa.gov/HYSPLIT.php), and the Global Data Assimilation System (GDAS). Additionally, we acknowledge the use of ISORROPIA II, accessible at (https://www.epfl.ch/labs/lapi/models-and-software/isorropia/), developed by the Schools of Earth & Atmospheric Sciences and Chemical & Biomolecular Engineering at the Georgia Institute of*

*Technology, for the calculation of aerosol pH and liquid water content.*"

**Comment (1):** In Table 1, aerosol pH values are with high standard deviations. Does it make the mean pH values less compatible? Are the data points actually quite overlapped?

**Response:**

As shown in Table 1, the standard deviations (SD) of pH value are approximately 0.8, indicating a considerable variability. This level of variability aligns with findings from a previous study, which documented similar SD values, exceeding 0.7 across various seasons (Ding et al., 2019). Ruan et al. (2022) observed SD values of 0.9 in clean air and 0.5 in heavily polluted air in Beijing, indicating that the observed SD levels of pH in our study are within expected ranges. In addition, the aerosol pH exhibited a pronounced difference between daytime and nighttime, as demonstrated by Figure R1. Although there is a substantial overlap in pH values between daytime and nighttime, this is anticipated due to the collection of aerosol samples under the same atmospheric conditions (either clean or slightly-polluted (SP) periods).

[Figure]

**Figure R1: Aerosol pH during clean and slightly-polluted (SP) periods. Boxes and error bars represent the 10th, 25th, 50th, 75th, and 90th percentiles from bottom to top, respectively.**

**References**

Ding, J., Zhao, P., Su, J., Dong, Q., Du, X., and Zhang, Y.: Aerosol pH and its driving factors in Beijing, Atmos. Chem. Phys., 19, 7939-7954, 10.5194/acp-19-7939-2019, 2019.

Ruan, X., Zhao, C., Zaveri, R. A., He, P., Wang, X., Shao, J., and Geng, L.: Simulations of aerosol pH

in China using WRF-Chem (v4.0): sensitivities of aerosol pH and its temporal variations during haze episodes, Geosci. Model Dev., 15, 6143-6164, 10.5194/gmd-15-6143-2022, 2022.

**Comment (2):** Figure 3, the data points are for both clean and slightly-polluted periods. They are mixed together, not separately marked like in Figure 5, why?

**Response:**

This is because we want to show the overall situation and demonstrate that the influence of aqueous-phase promoted acid processes on Fe solubility (%Fe$_S$) was profound not only during clean and SP periods, but also the whole sampling period. Figure 5 contains only Clean and SP data, with limited data points, so different markers are used for differentiation. In contrast, Figure 3 contains the whole sampling dataset, with about 140 points, making it visually unappealing and impractical to distinguish Clean and SP periods by using different markers (Figure R2), and also not conducive to performing regression analysis on clean and SP data points separately.

[Figure]

**Figure R2: Relationship between aerosol pH and the normalized relative abundance of main acidic species (2[SO$_4^{2-}$] + [NO$_3^-$]) with respect to the reconstructed PM$_{2.5R}$. The data of clean and SP periods is marked by triangles.**

Based on the above reasons, the dependence of pH on the ratio of (2[SO$_4^{2-}$] + [NO$_3^-$])/PM$_{2.5R}$ during clean and SP periods is shown in the subgraph at the bottom-left of Figure 3(a) show the robust influence of acid species on aerosol pH specifically during the clean and SP periods. In the revision,

we also added the dependence of pH on the ratio of ALWC/PM$_{2.5R}$ during clean and SP periods that is shown in the subgraph at the bottom-left of Figure 3(b). In terms of the relationship between %Fe$_S$ and the ratio of $(2[SO_4^{2-}] + [NO_3^-])$/PM$_{2.5R}$ during clean and SP periods, it was provided by Figure S4 in the ***Supplementary Information.*** We also modified the color-coding variable of ALWC in Figure 3(c) into unit aerosol mass (i.e., ALWC/PM$_{2.5R}$) to ensure consistency with Figure 3(a) and 3(b).

[Figure]

**Figure 3: Relationships among aerosol pH, the normalized relative abundance of ALWC (unit: μg m$^{-3}$) and main acidic species (= 2[SO$_4^{2-}$] + [NO$_3^-$], unit: μmol m$^{-3}$) with respect to the reconstructed PM$_{2.5}$ (PM$_{2.5R}$, unit: μg m$^{-3}$), and %Fe$_S$. The subgraph at the bottom-left of figures (a) and (b) show scatter plots during clean and SP periods with the linear regression line obtained by using the Igor Pro-based program developed by Wu and Yu (2018).**

[Figure]

**Figure S4: Same as Figure 3c in the manuscript but only for clean and SP periods.**

**Comment (3):** Figure 6, color bar is missing.

**Response:**

We added color bars in Figure 6 and more descriptions in the figure caption. The revised figure is shown as follows:

[Figure]

**Figure 6: RH-O$_x$ image plots colored by SOR during clean and SP periods. The last row and last column of the matrices represent the average value of SOR in the corresponding ranges of RH and O$_x$.**

---

## Author Comment (AC2)

*Point by Point Response to Review Comments*

**Daytime and nighttime aerosol soluble iron formation in clean and slightly-polluted moisture air in a coastal city in eastern China**
* * *
We thank the **Reviewer #2** for the detailed and constructive comments. We provide below point-by-point response to the comments. The reviewer's comments and the original contents of the manuscript are in **black**. The response text is in **blue**. Revisions in the manuscript are in **red**.

**General comments:**

The manuscript investigated the daytime and nighttime %$Fe_S$ in $PM_{2.5}$ under clean and SP conditions in a coastal city of China. They found that there was a significant difference between daytime and nighttime %$Fe_S$ under clean and SP conditions, respectively. Also, they explored the main factors influencing the dissolution of iron, such as aqueous-phase reactions vs. photochemical processes. Although the authors have explored the mechanism of iron dissolution in many ways, the paper suffered from many flaws. For the acidity of the aerosol, this study lacked both constraints of semivolatile gases and a scientific method to verify the accuracy of the simulation results, and thus the authors were very hasty in concluding that aerosols are more acidic, which is very uncritical. In addition, the authors reconstructed $PM_{2.5}$ based on aerosol chemical compositions, which was much higher than $PM_{2.5}$ at nearby station during the same period, is puzzling and suggested that the data in this study may be inaccurate. The authors characterized the relative strength of aerosol acidity, and the use of the $(2[SO_4^{2-}]+[NO_3^-])/PM_{2.5R}$ is incorrect, indicating that the author lacked basic knowledge. The language used in the text is still rough, and it is recommended that the authors strengthen the coherence of the language and simplify redundant expressions. In summary, I do not recommend this article for publication in EGUsphere at present form and it should be returned to the authors for major revision.

**General response:**

Thanks to the reviewer for the insightful comments. We have implemented the following significant

modifications:

1) We meticulously refined the linguistic expression of the manuscript to enhance precision and brevity.

2) We advanced the methodology for calculating aerosol pH by employing ISORROPIA II. By adopting a stringent condition outlined by Sun et al. (2018), we ensured the stability of our results and further validated the reliability of our aerosol pH calculations through comparisons between model simulations and field observations. We updated all sections pertaining to pH and Ambient Liquid Water Content (ALWC) within the manuscript.

3) We also compared the reconstructed $PM_{2.5}$ (i.e., $PM_{2.5R}$) with the $PM_{2.5}$ data from a nearby monitoring station (i.e., $PM_{2.5S}$), providing thorough explanations for the observed discrepancies between $PM_{2.5R}$ and $PM_{2.5S}$. Moreover, we elaborated the rationality of using the parameter of $(2[SO_4^{2-}] + [NO_3^-])/PM_{2.5R}$ in our work. For more detailed responses and explanations corresponding to the reviewer's comments, please refer to the detailed response to each comment below.

**Comment (1):** Page 2, line 31-34: Add references including more recent works. For example:

[1] Li, W., Xu, L., Liu, X., Zhang, J., Lin, Y., Yao, X., Gao, H., Zhang, D., Chen, J., and Wang, W., 2017. Air pollution–aerosol interactions produce more bioavailable iron for ocean ecosystems, Sci. Adv., 3, e1601749.

[2] Toner, B.M., 2023. An improved model of the ocean iron cycle. Nature, 620, 41-42.

**Response:**

We have added the references about recent works as the reviewer suggested on lines 32−35 and 40−42 as follows:

"Its deposition to high-nitrate, low-chlorophyll (HNLC) areas can stimulate phytoplankton boom, and ultimately enhance the absorption and fixation of atmospheric carbon in seawater (Watson et al., 1994; Watson and Lefévre, 1999; Toner, 2023). Studies have shown that only the soluble part of Fe ($Fe_S$) in aerosols is available to the phytoplankton, namely bioavailable Fe (Zhuang et al., 1992; Sugie et al., 2013; Li et al., 2017)."

"The %$Fe_S$ in primary particles can be significantly enlarged through atmospheric processes,

which is the consequence of aerosol acidification mainly via aqueous-phase reactions or absorption from the air (Solmon et al., 2009; Shi et al., 2015; Li et al., 2017; Hettiarachchi et al., 2019). ”

**Comment (2):** Page 2, line 31: It's not open oceans, exactly, but the HNLC regions.

**Response:**

We revised the expression on line 32 as follows:

"Its deposition to high-nitrate, low-chlorophyll (HNLC) regions can stimulate phytoplankton boom, and ultimately enhance the absorption and fixation of atmospheric carbon in seawater (Watson et al., 1994; Watson and Lefévre, 1999; Toner, 2023)."

**Comment (3):** Page 5, line 114: $NH_3$ is a major alkaline gas in the atmosphere that neutralizes the acidity of aerosols, and the lack of $NH_3$ will overestimate the acidity of aerosols. How does the author deal with this issue?

**Response:**

We completely agree with the reviewer that $NH_3$ plays a significant role in estimating aerosol acidity. In our previous calculation, we included $NH_3$ during the calculation in ISORROPIAII (version 2.3), utilizing the model in forward-mode. This means we considered both gas data (including $HNO_3(g)$, $HCl(g)$ and $NH_3(g)$) and aerosol data. The old manuscript line 114 outlined the method for $HNO_3(g)$ calculation, and the method for obtaining $HCl(g)$ and $NH_3(g)$ follows the procedure as that for $HNO_3(g)$. Addressing the reviewer's concerns regarding the reliability of deriving aerosol pH through only two iterations, we implemented a stringent criterion from Sun et al. (2018) to ensure result stability. Specifically, ISORROPIA was solved iteratively until the change in mass of the output $NO_3^-$ is below 1 %. We provide comprehensive explanations of how we determined the concentrations of $HNO_3(g)$, $HCl(g)$ and $NH_3(g)$ using this approach as follows:

1) Initially, the input of aerosol data was assumed as the sum of aerosol and gas data (specifically for $HNO_3$, $HCl$ and $NH_3$). This step yielded both gas and aerosol data from the first run of ISORROPIA.

2) In the second iteration, we added the gas data from the first run to the original aerosol data, and it was considered as the sum of gas data and aerosol data. This combination was treated as the new

input for calculating HNO₃(g), HCl(g) and NH₃(g), similar to the first run.

3) The same procedure was repeated for further iterations until the $NO_3^-$ output variation was less than 1% in mass.

Above calculation processes can be described by the following equations:

$$\text{Input}[C_{Aerosol} + C_{Gas}]_{N+1} = C_{Aerosol} + [C_{Gas}]_N$$

$$L = \left| \frac{[C_{NO_3^-}]_{N+1} - [C_{NO_3^-}]_N}{[C_{NO_3^-}]_N} \right| \times 100\%$$

where $C_{Aerosol}$ is the observed concentration of $NO_3^-$ (or $NH_4^+$, $Cl^-$); $C_{Gas}$ is the concentration of gaseous species of HNO₃(g) (or NH₃(g), HCl(g)); $[C_{Gas}]_N$ is the concentration of gaseous species of HNO₃(g) (or NH₃(g), HCl(g)) output by ISORROPIA in the $N^{th}$ run (N ≥ 1). The iteration was stopped until L < 1%.

After three iterations ($N_{max}$ = 3), we determined the newly calculated aerosol pH to be approximately 0.13 units lower than previously calculated. Significant correlations for pH ($r^2$ = 0.945) and ALWC ($r^2$ = 0.999) between the first and the fourth runs confirm the stability and reliability of the pH and ALWC estimations (Figure S2). The calculated NH₃(g) concentration was 2.1 ± 4.0 μg m⁻³, aligning with observations (mainly ranged from 0 to 8.0 μg m⁻³) reported by Chen et al. (2021) in Qingdao, 2019. To further validate the ISORROPIA results, we compared the simulated ions against measured values. As demonstrated in Figure S3, the significant correlations for $NO_3^-$ ($R^2$ = 0.625), $NH_4^+$ ($R^2$ = 0.982) and $Cl^-$ ($R^2$ = 0.521) underscore the high confidence level in the simulation outcomes.

[Figure]

**Figure S2: Relationships of aerosol pH and ALWC between the first run and the fourth run of ISORROPIA calculation.**

[Figure]

**Figure S3: Intercomparisons of simulated and measured concentrations of $NO_3^-$, $NH_4^+$ and $Cl^-$.**

The improved method for calculating aerosol pH utilizing ISORROPIA is detailed in Section 2.5 on lines 114−135:

"The concentrations of gaseous species (i.e., $NH_3(g)$, $HNO_3(g)$, $HCl(g)$) were not measured at the site. In alignment with the approach proposed by Sun et al. (2018), we devised a strategy to estimate the concentrations of these gaseous species. Initially, the input of aerosol data was assumed as the sum of aerosol and gas data (specifically for $HNO_3$, $HCl$ and $NH_3$). This step provided us with the first set of gas and aerosol data outputs. For the second run, the gas data output derived from the initial run was added to the original aerosol data, and it was considered as the sum of gas data and aerosol data just like the first run to calculate $HNO_3(g)$, $HCl(g)$ and $NH_3(g)$. The same method was employed for subsequent iterations until the variance in the $NO_3^-$ output below the 1% threshold in mass. The calculation processes can be described by the following equations:

$$\text{Input}[C_{Aerosol}+C_{Gas}]_{N+1}=C_{Aerosol}+[C_{Gas}]_N \tag{1}$$

R5

$$L = \left| \frac{[C_{NO_3^-}]_{N+1} - [C_{NO_3^-}]_N}{[C_{NO_3^-}]_N} \right| \times 100\% \tag{2}$$

where $C_{Aerosol}$ is the observed concentration of $NO_3^-$ (or $NH_4^+$, $Cl^-$), $C_{Gas}$ is the concentration of gaseous species of $HNO_3(g)$ (or $NH_3(g)$, $HCl(g)$), and $[C_{Gas}]_N$ is the concentration of gaseous species of $HNO_3(g)$ (or $NH_3(g)$, $HCl(g)$) output by ISORROPIA in the $N^{th}$ run ($N \geq 1$). The iteration was stopped until $L < 1\%$.

Finally, three times of iterations ($N_{max} = 3$) were determined when $L = 0.1\%$. The aerosol pH was calculated by using aqueous $H^+$ concentration and aerosol liquid water content (ALWC) outputted by ISORROPIA, as described by equation (3).

$$pH = -\log_{10} \frac{1000 \times H^+(aq)}{ALWC} \tag{3}$$

Significant correlations between the results of the first run and the fourth run were observed for pH ($r^2 = 0.95$) and ALWC ($r^2 = 0.99$), indicating the stability and reliability in estimating the pH and ALWC by ISORROPIA II (Figure S2). Moreover, the correlations of $NO_3^-$ ($r^2 = 0.71$), $NH_4^+$ ($r^2 = 0.98$) and $Cl^-$ ($r^2 = 0.51$) between the simulated results and measured concentrations are significant, demonstrating the robust confidence level of the simulated results (Figure S3)."


**Comment (6):** Line 131: Equation 2, The authors performed a mass reconstruction of $PM_{2.5}$, which needs to be compared with $PM_{2.5}$ from nearby monitoring stations to demonstrate the validity of this approach, at least in terms of trends. In addition, as can be seen in Table 1, the sum of the components of $PM_{2.5}$ at each stage exceeds the $PM_{2.5}$ concentration observed at the nearby ambient monitoring station, so why is there such a big difference? According to the data provided by the authors, the reconstructed $PM_{2.5}$ = 33.47 was calculated, which is almost twice as much as that of the nearby monitoring station. The difference is so obvious at such a close distance that the authors firstly need to ensure the accuracy of the data before the next analysis.

**Response:**

The comparison between the reconstructed $PM_{2.5}$ (i.e., $PM_{2.5R}$) with the $PM_{2.5}$ reported from an adjacent monitoring station (i.e., $PM_{2.5S}$) revealed congruent variation trends between these datasets, as depicted in Figure R3a, indicating the high confidence of the $PM_{2.5R}$ dataset.

However, we also noticed that the level of $PM_{2.5R}$ is higher than the $PM_{2.5S}$ observed from the nearby monitoring station. As shown in Figure R3b, the disparity between $PM_{2.5R}$ and $PM_{2.5S}$, expressed as the ratios of the difference to $PM_{2.5S}$ (i.e., $(PM_{2.5R} - PM_{2.5S})/PM_{2.5S}$), predominantly spans from 0.3 to 3.1, averaging at 1.7. This discrepancy is hypothesized to emanate predominantly from two reasons:

   1) The geographical disparity between these two observation locations. Figure S1 elucidates that the monitoring station (depicted as a yellow dot) is situated approximately 4.6 km to the southwest of the sampling site, nestled closer to the sea and further away from human activity areas. Therefore, the chosen standard monitoring station avoided the influence from anthropogenic activities to a large

extent, thereby accounting for the higher $PM_{2.5R}$ concentrations compared to $PM_{2.5S}$.

[Figure]

**Figure R3: Variations of PM$_{2.5}$ concentrations (a) and comparative parameters (b) during the sampling period. The x-axis shows the sample number. The black and red lines in (a) stand for the PM$_{2.5}$ concentrations obtained from a nearby monitoring station (i.e., PM$_{2.5S}$) and reconstructed PM$_{2.5}$ (i.e., PM$_{2.5R}$), respectively. The black line in (b) shows the ratios of the difference of PM$_{10S}$ and PM$_{2.5S}$ to PM$_{2.5S}$, indicating the degree of difference between PM$_{2.5}$ and PM$_{10}$. The red line in (b) shows the ratios of the difference of PM$_{2.5R}$ and PM$_{2.5S}$ to PM$_{2.5S}$, indicating the degree of difference between the reconstructed PM$_{2.5}$ and the PM$_{2.5}$ observed by the monitoring station.**

2) Different sampling instruments and the influence of coarse particles. Even though the capture efficiencies of the sampler cutting heads (PM$_{2.5}$ cyclone) for PM$_{2.5}$ of these two sites were same. i.e., 50%, the models and specifications of the instruments used at these two sites were different. This means that the geometric standard deviations of sampling efficiency may be different. The geometric standard deviation of sampling efficiency is usually expressed as the ratio of the particle aerodynamic diameter (D$_{a50}$) corresponding to a capture efficiency of 50% to the particle aerodynamic diameter (D$_{a84}$) corresponding to a capture efficiency of 84%. If the geometric standard deviations of sampling efficiency of two sampling instruments are different, then their capture efficiencies for coarse particles (e.g., PM$_{10}$) will be different. It may result in a noticeable difference in observed fine particles, because

the mass of atmospheric particulate matter is mainly concentrated in coarse particles. We compared the ratio of (PM$_{10S}$ - PM$_{2.5S}$)/PM$_{2.5S}$ with the ratio of (PM$_{2.5R}$ - PM$_{2.5S}$)/PM$_{2.5S}$ and found a high degree of consistency between them (Figure R3b). In other words, the higher the proportion of coarse particles in the atmosphere, the greater the difference between of PM$_{2.5R}$ and PM$_{2.5S}$. Significant differences usually occur when the proportion of coarse particles is high, mainly during the dust-influenced periods. But this study focused on clean and the SP periods, so absurd difference barely occur.

We added more explanations about this issue on lines 156−158:

"Because the nearby monitoring station is closer to the sea and less affected by human activities (yellow dot in Figure S1), the level of PM$_{2.5R}$ is higher than the observations from the monitoring station. But the trends of variations of these two datasets were consistent, indicating the high confidence of the PM$_{2.5R}$ dataset."

**Comment (7):** Line 158-160: Table 1, the calculated pHs are very low, indicating that the aerosols are strongly acidic. However, according to Wang et al., 2022, aerosols in the northern offshore are weakly acidic, and the pH of the aerosols in this study is much higher than that in the literature, which I think is most likely caused by the lack of data on NH$_3$, and therefore the pH calculated by the authors here lacks credibility. I suggest that the authors include data on NH$_3$ from the same period, or find data on atmospheric NH$_3$ concentration in Qingdao from previous literature.


Even though the aerosol pH values calculated by Wang et al. (2022) are higher than those observed in our study, it is important to note that the locations of sampling sites, seasons and years of respective studies varied significantly. Consequently, substantial differences in atmospheric conditions (e.g., relative humidity and temperature) and chemical compositions of aerosols are expected. Moreover, Figure R4 presented in the article by Wang et al. (2022) supports the conclusion that extremely low pH levels in coastal areas, such as Tanggu (with a pH approximated at 1.3) and the Bohai Sea (with a pH around 1.2), are plausible. This is attributed to the scarcity of alkaline substance sources (e.g., $NH_3$) over the oceanic regions (Zhou et al., 2018).

[Figure]

**Figure R4 Comparison of aerosol pH among various coastal and inland sites of China, which was reported by Wang et al. (2022).**

In our work, episodes of extreme aerosol pH were documented during the clean period. During the clean period, the air masses primarily originated from sea areas, leading to significantly acidic aerosol pH due to the absence of alkaline substance sources. In addition, throughout the entire

campaign, the calculated aerosol pH did not drop drastically, maintaining an average of around 2.5, including during clean, SP, heavily-polluted, fog-influenced, and dust-related periods.

On another note, the pronounced aerosol acidity observed in our work was mainly induced by the elevated proportions of $SO_4^{2-}$ and $NO_3^-$ in $PM_{2.5}$, resulting in a low level of ion balance (IB). The IB and equivalent ratios were calculated using the following equations, utilizing the charge-equivalent measured ion concentrations:

$$IB=[Anions]-[Cations]$$

$$[Anions]=\frac{[SO_4^{2-}]}{96}\times2+\frac{[NO_3^-]}{62}+\frac{[Cl^-]}{35.5}+\frac{[C_2O_4^{2-}]}{88}\times2+\frac{[F^-]}{19}$$

$$[Cations]=\frac{[Na^+]}{23}+\frac{[NH_4^+]}{18}+\frac{[K^+]}{39}+\frac{[Mg^{2+}]}{24}\times2+\frac{[Ca^{2+}]}{40}\times2$$

where [cations] and [anions] are the sum of charge-equivalent total molar concentrations ($\mu mol\ m^{-3}$) of cations and anions, respectively; $[Na^+]$, $[NH_4^+]$, $[K^+]$, $[Mg^{2+}]$, $[Ca^{2+}]$, $[SO_4^{2-}]$, $[NO_3^-]$, $[Cl^-]$, $[C_2O_4^{2-}]$ and $[F^-]$ are the mass concentrations ($\mu g\ m^{-3}$) of these ions in the atmosphere. In our work, IB mainly ranged from $-0.01\ \mu mol\ m^{-3}$ to $0.25\ \mu mol\ m^{-3}$ during the clean period and from $0.04\ \mu mol\ m^{-3}$ to $0.51$ $\mu mol\ m^{-3}$ during the SP period. The low IB facilitated enhanced $[H^+]$ levels, contributing to the low observed pH. These results align with the phenomena documented by Guo et al. (2015), where aerosol pH levels were typically below 1 in scenarios where the IB exceeded $0.05\ \mu mol\ m^{-3}$ (Figure R5). This can also support the rationality of our aerosol acidity.

[Figure]

**Figure R5 Comparison of ion balance with pH as provided by Guo et al. (2015).**

In the revision, we conducted more discussions on this issue on lines 215−222:

"The aerosol pH calculated in this work was evidently lower than many other areas of China (Liu et al., 2017; Wang et al., 2019; Xu et al., 2020). During the clean period, air masses mainly originated from the seas. Therefore, the aerosol pH can be very acidic because of the lack of sources of alkaline substances over the ocean, such as $NH_3$, $Ca^{2+}$, et al. (Zhou et al., 2018). Compared to the inland areas, much lower aerosol pH in coastal areas is reasonable (Wang et al., 2022). For instance, Zhou et al. (2018) reported that the pH of aerosols near the Bohai Sea can be as low as around 1.0. Moreover, they also found that the daytime aerosol acidity was significantly stronger than that during the nighttime in coastal areas. This observation aligns with the findings during clean periods in our study, which were characterized by the predominance of sea breezes."

**Response:**

In this study, the primary purpose of using the $PM_{2.5}$ reported from the monitoring station (i.e., $PM_{2.5S}$) was to classify cases into either 'clean' or 'SP' instances based on air quality. Due to the multitude of issues and difficulties associated with weighing high-volume sample collections, we did not perform mass concentration weighing and PM mass concentration calculation. To ensure that this classification criterion is comparable in other studies, and considering that reconstructed $PM_{2.5}$ (i.e., $PM_{2.5R}$) is only an estimated value of partial components with some uncertainties, we believe it is more appropriate to use $PM_{2.5S}$ data from national standard monitoring stations on lines 163−164 (old manuscript).

In terms of the contents on line 182 (old manuscript), the data presented were initially derived by using $PM_{2.5R}$. The proportions of chemical species in $PM_{2.5}$ in this study were calculated by dividing the concentration of chemical species by $PM_{2.5R}$. We have added further clarification in the manuscript on lines 158−160 to more clearly articulate this issue:

"In addition, any mention of ionic ratios or normalized parameters in the results and discussions of this paper indicates the data was divided by $PM_{2.5R}$."

**Comment (9):** Line 165: What are degraded air conditions? In general, atmospheric boundary layer

heights are higher during the day and lower at night, and here the authors' assertion that the higher $Fe_T$ and $Fe_S$ during the day are due to degraded atmospheric diffusion conditions lacks conclusive evidence. It is also possible that this is facilitated by stronger photochemical processes during the day.

**Response:**

The previous statement was not accurately phrased. The intention was to suggest that the elevated daytime values of $Fe_T$ and $Fe_S$ might be associated with higher aerosol pollution due to increased human activities. However, there isn't a significant difference between daytime and nighttime $PM_{2.5}$ values (16.9 vs. 16.4 μg m$^{-3}$). As the reviewer pointed out, the atmospheric boundary layer during the day is higher than at night, leading to more favorable dispersion conditions, which might contribute to the minimal diurnal variation in PM levels. Daytime human activities are more intense, and this too could be a reason for the higher daytime concentrations of both $Fe_T$ and $Fe_S$. Another possibility, as highlighted by the reviewer, is that daytime photochemical processes could lead to higher concentrations of $Fe_S$. Taking these points into consideration, we have reorganized the logic and made the following revision on lines 184–189:

"Compared to the nighttime, $Fe_T$ and $Fe_S$ concentrations were higher during the daytime, which were 289.2 ± 223.4 ng m$^{-3}$ and 20.0 ± 10.5 ng m$^{-3}$, respectively. Daytime levels of $Fe_T$ and $Fe_S$ were 1.5 times and 1.6 times as high as those observed at night, respectively. The increase in $Fe_T$ and $Fe_S$ during daytime may be linked to heightened human activities. Furthermore, the elevated $Fe_S$ during daytime could be attributed to photochemical processes, which promoted the dissolution of aerosol Fe, a topic to be discussed further in Section 4.2."

**Comment (10):** Line 187-190: The use of $(2[SO_4^{2-}]+[NO_3^-])/PM_{2.5R}$ to characterize the level of acidity in a unit of $PM_{2.5}$ is erroneous; the acidity of the aerosol, however, is determined by the amount of $H^+$ in the aqueous system, which is determined by the combination of acids and bases in the aerosol system.

**Response:**

We agree with the reviewers that $(2[SO_4^{2-}] + [NO_3^-])/PM_{2.5R}$ does not signify the acidity of aerosols. Instead, our use of this parameter aims to denote the proportion of acidic substances contained within a unit mass of particulate matter. Because sulfate and nitrate were dominant acidic species in WSIs

(constituting >75% of the mass) in this study and both of them are strong acids, the quantity of acidic substances in $PM_{2.5}$ can be evaluated through $(2[SO_4^{2-}] + [NO_3^-])/PM_{2.5R}$. Figure 3a proves that the significant dependence of pH on $(2[SO_4^{2-}]+[NO_3^-])/PM_{2.5R}$, elucidating that $SO_4^{2-}$ and $NO_3^-$ played dominant roles in driving aerosol pH. We also pinpointed possible mechanisms that caused this phenomenon in the manuscript. We have further clarified this issue on lines 222−224 as follows:

"In this study, we employed the ratio of acidic substances to PM, namely, $(2[SO_4^{2-}] + [NO_3^-])/PM_{2.5R}$, to characterize the level of acidic substances in a unit of $PM_{2.5}$, because $SO_4^{2-}$ and $NO_3^-$ were predominant acidic species within WSIs (>75% in mass)."

The dependence of pH on $(2[SO_4^{2-}] + [NO_3^-] - [NH_4^+])/PM_{2.5R}$ was also studied as shown in Figure R6. The dependence of pH on $(2[SO_4^{2-}] + [NO_3^-] - [NH_4^+])/PM_{2.5R}$ was far less significant than on $(2[SO_4^{2-}] + [NO_3^-])/PM_{2.5R}$, indicating the critical roles of $SO_4^{2-}$ and $NO_3^-$ in regulating the aerosol pH.

[Figure]

**Figure R6: Dependence aerosol pH on $(2[SO_4^{2-}] + [NO_3^-])/PM_{2.5R}$ (left) and $(2[SO_4^{2-}] + [NO_3^-] - [NH_4^+])/PM_{2.5R}$ (right) during the whole sampling period. Circles are colored by ALWC/$PM_{2.5R}$ (unit: µg µg$^{-1}$).**

**Comment (11):** Line 199-203: Why are the authors' calculated daytime and nighttime aerosol pH values so large for different pollution scenarios when the SNA percentages are close? During the cleaning period, daytime pH is lower than nighttime, and during the pollution period is nighttime lower than daytime?

**Response:**

We explained this issue by using the relationships between pH and $(2[SO_4^{2-}] + [NO_3^-])/PM_{2.5R}$ on lines 256–260 in Section 4.1:

"Especially during clean and SP periods (r = 0.62, Figure 3a), the slope of the regression line was approximately –602.99, indicating that a variation of 1.0 nmol $\mu g^{-1}$ of the acidic species content in $PM_{2.5}$ can lead to a noticeable fluctuation of aerosol pH (about 0.60). For instance, the daytime aerosol pH was 0.60 lower than that of the nighttime during the clean period, even though the difference of the two acidic species content was only about 1.0 nmol $\mu g^{-1}$."

Actually, a variation of 1.0 nmol $\mu g^{-1}$ of $(2[SO_4^{2-}]+[NO_3^-])/PM_{2.5R}$ is far from being minor. We evaluated the pH variation, i.e., $\Delta$pH, caused by a fluctuation of 1.0 nmol $\mu g^{-1}$ of $(2[SO_4^{2-}]+[NO_3^-])/PM_{2.5R}$. Assuming the $NH_4^+$ concentration is stable, an increase of $(2[SO_4^{2-}]+[NO_3^-])/PM_{2.5R}$ of 1.0 nmol $\mu g^{-1}$ indicates an $[H^+]$ increase of 1.0 nmol $\mu g^{-1}$. The resulting pH variation can be calculated by the following equation:

$$\Delta pH = pH_{original} - pH_{present} = lg[H^+]_{present} - lg[H^+]_{original}$$

$$= lg\frac{n(H^+)_{original} + \Delta n(H^+)}{ALWC_{original}} - lg\frac{n(H^+)_{original}}{ALWC_{original}}$$

$$= lg\left(\frac{n(H^+)_{original} + \Delta n(H^+)}{ALWC_{original}} \times \frac{ALWC_{original}}{n(H^+)_{original}}\right) = lg\frac{n(H^+)_{original} + \Delta n(H^+)}{n(H^+)_{original}}$$

$$= lg\left(1 + \frac{\Delta n(H^+)}{n(H^+)_{original}}\right) \qquad (1)$$

where n($H^+$) is the molar concentration of $H^+$ in the air; $\Delta n(H^+)$ is the variation of n($H^+$). Considering that the $PM_{2.5}$ concentration was about 16.5 $\mu g\ m^{-3}$ during the clean period, $\Delta n(H^+)$ can be evaluated as 16.5 $\mu g\ m^{-3}$ multiplied by 1.0 nmol $\mu g^{-1}$, i.e., $16.5 \times 10^{-9}$ mol $m^{-3}$. Assuming that the aerosol pH during the clean period is 1.0 and ALWC is 40.0 $\mu g\ m^{-3}$ based on our observations, we can obtain that:

$$pH_{original} = -lg\frac{n(H^+)_{original}}{V(ALWC_{original})} = -lg\frac{n(H^+)_{original}}{\frac{40\ \mu g \cdot m^{-3}}{10^3\ kg \cdot m^{-3}}} = -lg\frac{n(H^+)_{original}}{4 \times 10^{-8}\ L \cdot m^{-3}} = 1.0$$

So, $n(H^+)_{original}$ is about $4 \times 10^{-9}$ mol $m^{-3}$. According to Equation (1), $\Delta$pH = lg(1+16.5/4) = 0.71. Therefore, an increase of 1.0 nmol $\mu g^{-1}$ of $(2[SO_4^{2-}]+[NO_3^-])/PM_{2.5R}$ can cause a significant pH

variation of about 0.71. Thus, even though the difference of $(2[SO_4^{2-}]+[NO_3^-])/PM_{2.5R}$ levels between daytime and nighttime was visually small during clean (0.0010 μmol μg) and SP periods (0.0005 μmol μg), aerosol pH was very sensitive to it.

**Comment (12):** Line 255-256: "A decrease of 10% in RH resulted in a notable reduction of 7.6% in SOR and 7.2% in NOR (Figure 5)." How can the authors come to such a solid conclusion when the correlation here is not very high? What is the level of significance?

**Response:**

We calculated the level of significance about the correlations between RH and SOR and NOR. As shown in Figure 5, SOR was significantly correlated with RH when RH < 78% (R = 0.64, p < 0.01). NOR was also dependent on RH when RH < 75% (R = 0.46), but the level of significance was low (p > 0.05). So, we revised the relevant contents on lines 25−27 and 282−291 in the manuscript as follows:

"Furthermore, the oxidation rates of sulfur (SOR)  displayed a strong correlation with RH, particularly when RH was below 75%. A 10% increase in RH corresponded to a 7.6% rise in SOR , which served as the primary driver of the higher aerosol acidity and %Fe$_S$ at night."

"RH is a key factor in the formation of $SO_4^{2-}$ and $NO_3^-$ through heterogeneous/aqueous-phase reactions within aerosols (Wang et al., 2016; Liu et al., 2020; Hou et al., 2022). As demonstrated in Figure 5, the strong dependency of the oxidation rate of sulfur (SOR, defined as $[SO_4^{2-}]/([SO_4^{2-}] + [SO_2])$) on RH was observed under moderate humid conditions (r = 0.64, p < 0.01). But the nitrogen (NOR, defined as $[NO_3^-]/([NO_3^-] + [NO_2])$) had a poor dependence on RH (r = 0.46, p > 0.05). A decrease of 10% in RH resulted in a notable reduction of 7.6% in SOR (Figure 5). Such a striking RH dependence was observed mainly during the SP period, indicating the significant role of heterogeneous reactions in controlling the formation of $SO_4^{2-}$. Therefore, the facilitation of aqueous-phase conversions leading to the formation of $SO_4^{2-}$ was more pronounced at night during the SP period, attributed to the high RH. This, in turn, resulted in a high proportion of $SO_4^{2-}$ and acidic species, as well as the elevated SOR (Table 1, Figures 2b and S5). The nighttime aerosol pH was approximately 0.18 units lower than that during daytime, but this slight variation did not hinder the efficient formation of Fe$_S$ during nighttime in SP periods."

[Figure]

**Figure 5: The dependence of SOR (a) and NOR (b) on RH during clean and slightly-polluted periods. The fitting of the regression line between SOR and RH was fitted when RH<78%. The fitting of the regression line between NOR and RH was fitted when RH<75% and one deviation point (the red circle in (b)) was removed.**

**Comment (13):** Line 297: Figure 6 confused me. What are the different colors in the figure represent?

**Response:**

The color in Figure 6 stands for the level of SOR in a certain range of RH and $O_X$. We added color bars in Figure 6 and more descriptions in the figure caption. The revised figure is shown as follows:

[Figure]

**Figure 6: RH-$O_x$ image plots colored by SOR during clean and SP periods. The last row and last column of the matrices represent the average value of SOR in the corresponding ranges of RH and $O_x$.**

**Comment (14):** Line 316: For Figure 7, I believe the effect of oxalic acid on iron dissolution is important enough to suggest that the authors devote a chapter to the mechanism of oxalic acid and iron dissolution. In the current version, the authors did not do a good job of exploring the facilitating effect of oxalic acid on iron dissolution.

**Response:**

We devoted a chapter (Section 4.2.2) to the mechanisms of iron dissolution by oxalate promotion now, on lines 356−403 as follows:

**4.2.2 The enhancement of %$Fe_S$ promoted by oxalate-related conversions**

[revised manuscript text omitted]

---

## Author Response (AR2)

**Point by Point Response to Review Comments**

**Daytime and nighttime aerosol soluble iron formation in clean and slightly-polluted moisture air in a coastal city in eastern China**
* * *
We thank the **Reviewer #2** for the detailed and constructive comments. We provide below point-by-point response to the comments. The reviewer's comments and the original contents of the manuscript are in **black**. The response text is in **blue**. Revisions in the manuscript are in **red**.

**Comment (1):** Page R2: it is "Aerosol liquid water content (ALWC)" not "Ambient Liquid Water Content (ALWC)" in General response.

**Response:**

Many thanks for the reviewer's corrections.

**Comment (2):** Page R3: Please give a link to ISORROPIA II (version 2.3) in the text so that readers can learn more about this model.

**Response:**

We have provided the link to ISORROPIA II on line 109 of the manuscript.

"ISORROPIA thermodynamic equilibrium model (version II, https://www.epfl.ch/labs/lapi/models-and-software/isorropia/iso-code-repository/) was employed to estimate gas concentrations and aerosol water pH (Song et al., 2018)."

**Comment (3):** Page R4: "As demonstrated in Figure S3, the significant correlations for $NO_3^-$ ($R^2$ = 0.625), $NH_4^+$ ($R^2$ = 0.982) and $Cl^-$ ($R^2$ = 0.521) underscore the high confidence level in the simulation outcomes." Here, the regression coefficients in the text do not match those in the figures, and the authors are advised to correct the data if it was used in the manuscript. For example, the $R^2$ for $NO_3^-$ and $Cl^-$ are 0.625 and 0.521, respectively, but are 0.705 and 0.511 in the graph. Moreover, the

intercomparisons of simulated and measured concentrations of $NO_3^-$, $NH_4^+$ and $Cl^-$ in R5 can be moved to the supplementary materials.

**Response:**

We are sorry for the mistakes in the response contents. But the correlations ($r^2$) in Figure S3 and the manuscript are right, i.e., the correlations of $NO_3^-$, $NH_4^+$ and $Cl^-$ between the simulated results and measured concentrations are 0.71, 0.98, 0.51, respectively. Moreover, Figures S2 and S3 in Pages R4 and R5 have already added in the supplementary materials.